# Toggle switch residues control allosteric transitions in bacterial adhesins by participating in a concerted repacking of the protein core

**Dagmara I. Kisiela**[1☉¤a], **Pearl Magala**[2☉], **Gianluca Interlandi**[3], **Laura A. Carlucci**[3], **Angelo Ramos**[2], **Veronika Tchesnokova**[1], **Benjamin Basanta**[2,4¤b], **Vladimir Yarov-Yarovoy**[5], **Hovhannes Avagyan**[1], **Anahit Hovhannisyan**[1], **Wendy E. Thomas**[3], **Ronald E. Stenkamp**[2,6], **Rachel E. Klevit**[2]\*, **Evgeni V. Sokurenko**[1]\*

1 Department of Microbiology, University of Washington, Seattle, Washington, United States of America,
2 Department of Biochemistry, University of Washington, Seattle, Washington, United States of America,
3 Department of Bioengineering, University of Washington, Seattle, Washington, United States of America,
4 Institute for Protein Design, University of Washington, Seattle, Washington, United States of America,
5 Department of Physiology and Membrane Biology, University of California, Davis, California, United States of America, 6 Department of Biological Structure, University of Washington, Seattle, Washington, United States of America

☉ These authors contributed equally to this work.
¤a Current address: Codexis, Inc., Redwood City, California, United States of America
¤b Current address: Department of Integrative Structural and Computational Biology, The Scripps Research Institute, La Jolla, California, United States of America
\* klevit@uw.edu (REK); evs@uw.edu (EVS)

**Data Availability Statement:** All relevant data are within the manuscript and its Supporting Information files.

## Abstract

Critical molecular events that control conformational transitions in most allosteric proteins are ill-defined. The mannose-specific FimH protein of *Escherichia coli* is a prototypic bacterial adhesin that switches from an 'inactive' low-affinity state (LAS) to an 'active' high-affinity state (HAS) conformation allosterically upon mannose binding and mediates shear-dependent catch bond adhesion. Here we identify a novel type of antibody that acts as a kinetic trap and prevents the transition between conformations in both directions. Disruption of the allosteric transitions significantly slows FimH's ability to associate with mannose and blocks bacterial adhesion under dynamic conditions. FimH residues critical for antibody binding form a compact epitope that is located away from the mannose-binding pocket and is structurally conserved in both states. A larger antibody-FimH contact area is identified by NMR and contains residues Leu-34 and Val-35 that move between core-buried and surface-exposed orientations in opposing directions during the transition. Replacement of Leu-34 with a charged glutamic acid stabilizes FimH in the LAS conformation and replacement of Val-35 with glutamic acid traps FimH in the HAS conformation. The antibody is unable to trap the conformations if Leu-34 and Val-35 are replaced with a less bulky alanine. We propose that these residues act as molecular toggle switches and that the bound antibody imposes a steric block to their reorientation in either direction, thereby restricting concerted repacking of side chains that must occur to enable the conformational transition. Residues homologous to the FimH toggle switches are highly conserved across a diverse family of

**Funding:** This work was supported by National Institutes of Health Grants R01AI119675 (WET, REK and EVS), R21AI147575 (EVS) and T32GM008268 (LAC). The funders had no role in study design, data collection and analysis, decision to publish, or preparation of the manuscript.

fimbrial adhesins. Replacement of predicted switch residues reveals that another *E. coli* adhesin, galactose-specific FmlH, is allosteric and can shift from an inactive to an active state. Our study shows that allosteric transitions in bacterial adhesins depend on toggle switch residues and that an antibody that blocks the switch effectively disables adhesive protein function.

## Author summary

To bind their ligands, allosteric proteins shift between 'inactive' and 'active' states, but molecular details of the conformational changes during the transition are often unclear. We describe a monoclonal antibody against the mannose-specific bacterial adhesin, FimH, that blocks the conformational transition in both directions. The antibody-trapped LAS and HAS conformations of FimH are unable to mediate bacterial adhesion under dynamic shear conditions. We propose that the conformational trapping involves a steric block of the core-to-surface switching of certain residues which is critical for the allosteric transitions. Furthermore, we demonstrate that the allosteric switches are structurally and functionally conserved across a broad spectrum of bacterial fimbrial adhesins.

## Introduction

Receptor-ligand interactions are essential for cell adhesion, molecular transport, signal transduction, enzymatic catalysis, etc. Most ligand-binding proteins adopt both 'inactive' and 'active' conformations and the transition between the two states is usually allosterically controlled. Inactive states are characterized by low-affinity ligand binding, while active states have much higher affinity towards the ligand [1–3]. The conformational kinetics of the allosteric states and the dynamic mechanisms of ligand recognition have been difficult to decipher due to a lack of tools that can block the transitions without favoring one state over the other.

FimH protein is the most prevalent adhesin of *Escherichia coli*: over 90% of strains express FimH under one or another condition [4]. FimH is a prototypic allosteric receptor protein that is positioned on the tip of hair-like fimbriae (or pili) and binds to mannosylated cell surface glycoproteins [5]. The protein is composed of two domains: an N-terminal lectin domain (LD) contains the mannose-binding pocket and a C-terminal pilin domain (PD) anchors the adhesin to the fimbriae [6]. The LD can assume two end-point conformational states–a low-affinity state (LAS) and a high-affinity state (HAS)–and there is evidence for at least one intermediate state (Fig 1A). The LAS has a compressed β-sandwich structure and an open mannose-binding pocket. Inter-domain interactions with the PD at the opposite end of the LD from the binding pocket stabilize the LAS [7]. In contrast, the binding pocket of the HAS is more closed, the β-sandwich is more elongated, and LD/PD contacts are disrupted [6]. Ligand binding to LAS FimH induces conformational changes to an intermediate state in which the binding pocket is closed but the interdomain interaction is preserved [8] and the end-point HAS in which the LD and PD are separated is achieved via an allosteric transition [9–11]. Importantly, the domain separation and, therefore, transition from the intermediate to HAS is both strongly favored and sustained by mechanical tensile force. Force-induced domain separation may occur on a faster timescale than the mannose-induced separation and is critical during FimH-mediated bacterial adhesion under physiological shear induced by flow conditions. Thus, FimH is able to form a so-called catch bond with its ligand, in which the lifetime

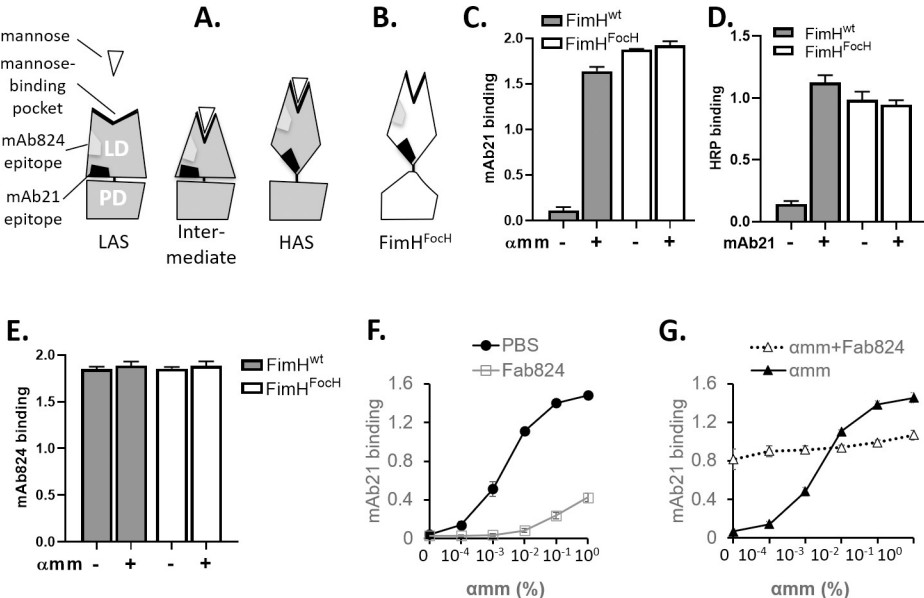

**Fig 1. Recognition of FimH variants by mAb21 and mAb824.** Cartoon representation of LAS, intermediate and HAS conformations (A) and FimH$^{FocH}$ mutant variant (B) of FimH adhesin. The putative locations of mAb21 and mAb824 epitopes are indicated as black and grey shapes, respectively. (C) Binding of mAb21 in the absence or presence of 1% αmm to plate-coated fimbriae carrying different variants of FimH adhesin. (D) HRP binding to plate-coated fimbriae in the absence of mAb21 and upon forming a complex with mAb21 in the presence of mannose; mean ± SD, n = 1. (E) Binding of mAb824 in the absence or presence of 1% αmm to plate-coated fimbriae carrying different variants of FimH adhesin. (F-G) Binding of mAb21 to plate-coated FimH$^{wt}$ fimbriae in the presence of different concentration of αmm; and upon no treatment (solid line with closed circle) or pre-treatment with αmm (solid line with closed triangle), Fab824 (solid gray line with open square), and Fab824 and αmm together (dashed line with open triangle), as designated in the figure Data (C and E-F) are mean absorbance values at 650 nm ± SEM, n = 3.

of binding increases under shear [9,10,12]. The mannose-induced transition may occur on a slower timescale but likely plays a critical role during adhesion under low-shear, static, or equilibrium conditions. Removal of the PD to create an isolated LD or structural alterations that lead to disruption of the interdomain interface in full-length FimH both strongly favor the HAS conformation of LD [7,9]. However, while the structural differences between the endpoint LAS and HAS of FimH are well defined, the key molecular events that underlie the allosteric transition remain undefined.

Conformation-specific monoclonal antibodies are powerful tools for studying allosteric receptor proteins. HAS-specific monoclonal antibodies have been identified for FimH, with the representative antibody mAb21 binding to the interdomain region of LD that is exposed in the HAS [10,13]. To date, no antibody has been identified that blocks the conformational transition between LAS and HAS in both directions. Here we characterize a novel monoclonal antibody that recognizes both LAS and HAS conformations of FimH. Binding to either state of the adhesin inhibits the allosteric transition in both directions. We propose that antibody binding blocks the ability of particular residues within the LD-antibody interface to switch between solvent-exposed and core-buried orientations, a step we show to be critical for the allosteric transition.

## Results

### mAb824 equally recognizes alternative conformational states of FimH

We compared binding of two monoclonal antibodies with two forms of FimH in purified fimbriae. The two antibodies are the previously characterized mAb21 and a novel mAb824, which

is the focus of this study. One form of FimH was 'wild-type' FimH adhesin (FimH$^{wt}$) derived from *E. coli* K12 strain that is structurally identical to the FimH found commonly in uropathogenic *E. coli*, including a model cystitis strain J96. The other form was a recombinant FimH variant (FimH$^{FocH}$) that contains a 15-residue substitution in its PD that alters the interdomain interface, based on a different *E. coli* adhesin, FocH [9]. In the absence of ligand, the LD of FimH$^{wt}$ adopts the open-pocket LAS conformation that is not recognized by mAb21 (Fig 1A and 1C). Binding of α-methyl-D-mannopyranoside (αmm) causes the transition from LAS to HAS, thereby enabling mAb21 to recognize FimH$^{wt}$ (Fig 1A and 1C). mAb21 binding to the αmm-induced HAS stabilizes the latter and, upon washing out αmm, enables FimH$^{wt}$ fimbriae to strongly bind soluble horseradish peroxidase (HRP), a heavily mannosylated glycoprotein that binds very weakly to FimH$^{wt}$ under static equilibrium conditions, i.e. the LAS conformation (Fig 1D). Consistent with its disrupted LD/PD interface [9], the LD in FimH$^{FocH}$ is predominantly in the HAS (Fig 1B), based on previous studies showing that FimH$^{FocH}$ has similar binding affinities as purified isolated LD, which is constitutively in HAS [9,10]. As predicted, FimH$^{FocH}$ fimbriae bind mAb21 equally well in the absence and presence of αmm and their binding to HRP is equally strong before or after forming a complex with mAb21 (Fig 1C and 1D).

Unlike mAb21, mAb824 binds to both FimH$^{wt}$ and FimH$^{FocH}$ in the absence or presence of ligand (Figs 1E and S1A), implying that its epitope is intact in both LD conformations and does not overlap with the mannose-binding pocket. Furthermore, mAb21 and mAb824 antibodies can bind to the HAS simultaneously (S1B Fig), indicating that the two epitopes do not overlap. Together these observations imply that the mAb824 epitope is located somewhere between the ligand-binding pocket and the interdomain interface (Fig 1A). The kinetics of mAb824 binding in the absence and presence of αmm are remarkably similar, with similar association rates ($k_{on}$ of 2.6 ± .005 and 2.4 ± .007 x 10$^5$ M$^{-1}$s$^{-1}$, respectively) and immeasurably low dissociation rates (S1C Fig). Altogether, the binding measurements are consistent with the novel mAb824 recognizing both the LAS and HAS conformations of FimH.

## mAb824 blocks the allosteric transition of FimH in either direction

To assess if mAb824 binding to one conformation affects the ability to transition to the other, binding experiments were carried out on FimH$^{wt}$ pre-treated with the Fab fragment of mAb824 (Fab824), when the LD is in the LAS or, alternatively, HAS conformation. Again, mAb21 binding was used as a probe for the HAS.

In the case of FimH$^{wt}$ pre-treated with PBS only, increasing the αmm concentration generates mAb21 binding in a dose-dependent manner, with a plateau reached between 0.1–1.0% αmm ("PBS", Fig 1F). FimH$^{wt}$ pre-treated with Fab824 (presumably forming the LAS/mAb824 complex) shows significantly diminished binding to mAb21, requiring approximately three orders of magnitude higher αmm to exhibit a measurable mAb21 binding and, at 1% αmm, demonstrating only ~30% of binding relatively to the PBS-treated sample. Thus, mAb824 appears to inhibit the LAS-to-HAS transition induced by αmm. Incubation of Fab824 with FimH$^{wt}$ after the latter is activated by αmm (at 1% for 30 min) should produce an HAS/mAb824 complex. The pre-treatment with mAb824 in the presence of αmm (followed by a washout step to remove αmm) yields a species that binds mAb21 at ~70% of maximal level even without αmm and shows no αmm-dependence of further activation (open triangles, Fig 1G). This is in contrast to FimH$^{wt}$ pre-incubated with αmm but without Fab824 that fully returned to LAS upon wash out of αmm (closed triangles, Fig 1G). Taken together, the observations indicate that once bound to mAb824, the LAS/HAS transition of FimH$^{wt}$ is strongly inhibited in both directions. In other words, mAb824 effectively traps FimH in the conformation to which it initially bound.

## Blocking the allosteric transitions inhibits association of FimH with the ligands

We tested whether blocking the allosteric transitions by mAb824 affects ligand binding kinetics of FimH$^{wt}$ and FimH$^{FocH}$. Interaction of immobilized fimbriae with bovine RNase B, a model FimH ligand rich in N-linked high-mannose (Man$_5$) oligosaccharides, was measured by bio-layer interferometry (BLI) analysis. FimH$^{wt}$ demonstrated nanomolar affinity (149 ± 4 nM), with an association rate constant $k_{on}$ of 1.1 x 10$^6$ M$^{-1}$s$^{-1}$ and a dissociation rate constant $k_{off}$ of 0.26 s$^{-1}$ (Table 1 and S2A Fig). Binding to FimH$^{FocH}$ fimbriae occurred with ~50-fold slower association rate but also ~1000-fold slower dissociation rate compared to FimH$^{wt}$, for a ~10-fold increase in binding affinity (Table 1 and S2A and S2B Fig). Pre-treatment of FimH$^{wt}$ with Fab824 resulted in a large decrease in the RNase B-binding signal to a level as low as in the presence of saturating concentrations of soluble αmm. This decrease did not allow for reliable measurement of $k_{on}$ and $k_{off}$ (S2A Fig). Fab824 treatment of FimH$^{FocH}$ resulted in a slight increase in the dissociation rate and more than 3-fold drop in the association rate, leading to a ~5-fold decrease in the binding affinity based on $K_D$ (Table 1 and S2B Fig).

We also tested whether blocking the allosteric transitions affects cell-adhesive properties of FimH under shear conditions using the classical test of guinea pig red blood cell (RBC) aggregation by bacteria under dynamic rocking conditions. Bacteria expressing FimH$^{wt}$, which is able to shift from LAS to HAS under shear conditions, readily aggregates RBC within 2 minutes of rocking (Table 1). Bacteria expressing the HAS-trapped FimH$^{FocH}$ required longer times and higher doses to aggregate RBC and the aggregates produced were smaller even at the highest bacteria dose (Table 1 and S3 Fig). These observations are consistent with the lower binding on-rate of the HAS conformation. Pretreatment with mAb824 completely abrogated aggregation caused by bacteria expressing either FimH$^{wt}$ or FimH$^{FocH}$ (Table 1).

**Table 1.** Binding properties of FimH$^{wt}$ and FimH$^{FocH}$.

| TEST | FimH variant | | PBS | αmm | mAb824* |
|---|---|---|---|---|---|
| BLI‡ | FimH$^{wt}$ | $k_{on}$ (M$^{-1}$s$^{-1}$) | 110 ± 3.2 x 10$^4$ | Not calculated | Not calculated |
| | | $k_{off}$ (s$^{-1}$) | 256 ± 6 x 10$^{-3}$ | | |
| | | $k_{act}$ (s$^{-1}$) | 7.2 ± 0.1 x 10$^{-3}$ | | |
| | | $k_{inact}$ (s$^{-1}$) | 12.8 ± 0.1 x 10$^{-3}$ | | |
| | | $K_D$ (nM)$^♯$ | 149 ± 4 | | |
| | FimH$^{FocH}$ | $k_{on}$* (M$^{-1}$s$^{-1}$) | 2.2 ± .01 x 10$^4$ | Not tested | 0.71 ± .002 x 10$^4$ |
| | | $k_{off}$* (s$^{-1}$) | 0.31 ± .001 x 10$^{-3}$ | | 0.47 ± .001 x 10$^{-3}$ |
| | | $K_D$ (nM)** | 13.9 ± .007 | | 66.6 ± .02 |
| RBC agglutination$^†$ | FimH$^{wt}$ | | 2 min | > 60 min | > 60 min |
| | FimH$^{FocH}$ | | 10 min | > 60 min | > 60 min |

\* Fab824 and mAb824 were used in the BLI and RBC agglutination test, respectively

† Time (in min) upon which noticeable aggregation of RBC was observed

‡ Kinetic parameters for FimH$^{wt}$ and FimH$^{FocH}$ fimbriae binding to RNase B determined for BLI data presented in S2A and S2B Fig. As described previously [32], FimH$^{wt}$ displays complex kinetics that require a conformational change model with four rate constants: the LAS association rate ($k_{on}$), the LAS dissociation rate ($k_{off}$), rate of transition from LAS to HAS ('activation' rate; $k_{act}$), and rate of transition from HAS to LAS ('inactivation' rate; $k_{inact}$). In contrast, FocH displays simple kinetics that can be fit with a 1:1 binding model with two parameters for the HAS association rate ($k_{on}$*), and the HAS dissociation rate ($k_{off}$*). The table shows the values calculated for data from one representative BLI experiment

♯ $K_D$ value was determined according to the conformational change model: $K_D = \frac{k_{off}}{k_{on}} \left( \frac{k_{inact}}{k_{inact} + k_{act}} \right)$

\** $K_D$ value was determined according to the 1:1 binding model: $K_D = \frac{k_{off}*}{k_{on}*}$

In sum, the mAb824-rendered inability of the LAS to convert to the HAS yields an LD that is unable to associate even a highly mannosylated ligand at a measurable level. Furthermore, the on-rate of the antibody-trapped HAS is markedly reduced relative to the non-trapped FimH[FocH]. Both effects could underlie the complete inhibition of bacterial binding to target cells under dynamic shear conditions observed here.

## mAb824 functional epitope in LD is conformationally conserved in LAS and HAS

In an attempt to gain insight into the molecular mechanism of mAb824's effect on FimH, its epitope was mapped by mutagenesis. Analysis of a library of 168 point mutations covering 83 LD positions revealed that substitution of S80 and Y82 with alanine eliminated or significantly decreased mAb824 binding to both FimH[wt] and FimH[FocH] (S1 Table), identifying those residues as energetically critical for the antibody binding and, therefore, part of the 'functional epitope'[14,15]. Other substitutions that severely affected mAb824 binding were of G79, S81, and P91 with bulkier arginine. These residues are proximal to S80 and Y82 (Fig 2A). Therefore, as predicted above (Fig 1A), the mAb824 epitope is located between the mannose-binding pocket and LD-PD interface, with the shortest Cα distance from epitope residues (P91 and S81) in FimH[wt] to mannose-binding pocket (Q133) and interdomain interface (V30) residues is 11.7 Å and 14 Å, respectively (measured on PDB 4xo9) (Fig 2A). Notably, all five mutagenesis-sensitive residues exist in similar structural positions in the two conformations: neither their

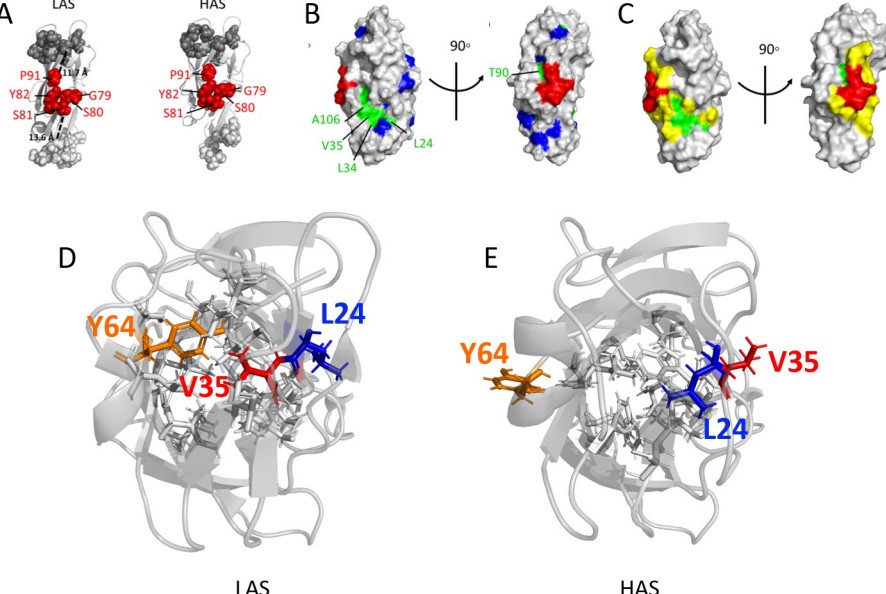

**Fig 2. Epitope of mAb824 mapped on the FimH lectin domain structure.** (A) Views of crystal structures of the FimH LD in the LAS (PDB code 4xo9) and HAS (PDB code 4xo8) conformations. Residues whose mutation affected mAb824 binding are labeled and shown as red spheres. Ligand-binding pocket residues on the "top" of the lectin domain are shown as dark grey spheres and the interdomain region residues on the "bottom" are shown as light grey spheres. (B) Surface rendition of LD (PDB code 3zpd) showing residues with large perturbations in the NMR spectrum upon Fab824 binding (green). Methyl residues that do not exhibit large perturbations are shown in dark blue. The mAb824 epitope determined by mutagenesis is shown in red. (C) The putative structural epitope of mAb824 mapped on an LD HAS structure (PDB code 3zpd). Residues immediately bordering the functional epitope (red) and the NMR-determined patch (green) are marked in yellow (N23, N33, D37, T74, K76, S78, P83, T86, E89, R92, V93, V105, Y108). (D) The V35/L34 and Y64 solvent-core switches. Ribbon representation of FimH lectin domain in LAS (PDB code 4xo9) and HAS (PDB code 4xo8) with the alternate orientations of V35 (red stick) and L34 (blue). Y64 (described later in study) is in orange sticks.

solvent accessible surface area (SASA) (S2 Table) nor their Cα positions (RMSD of 1.1 ± 0.3 Å) differ substantially.

The mutagenesis identifies G79-Y82 and P91 as the functional epitope of the mAb824-LD interaction interface, with a combined SASA of 326 Å$^2$ (measured on PDB 4xo9). This area is much smaller than expected for a total antigen-antibody contact area (aka 'structural epitope') which generally expands beyond the binding-critical functional epitope and can include surface residues that are energetically not critical for antibody binding [14–16]. Analyses of crystal structures of numerous antigen-antibody complexes reveal that such areas within a protein antigen are generally composed of 15–25 solvent-exposed residues and cover 1100 ± 244 Å$^2$ on average and 600–650 Å$^2$ at a minimum [14–16].

## NMR spectroscopy detects the mAb824-perturbed residues within the putative antibody contact area

We used NMR spectroscopy on purified LD in complex with Fab824 to map the antibody contact area. At 69 kDa, the LD-Fab824 complex is not amenable to standard ($^1$H, $^{15}$N)-HSQC experiments, but methyl groups yield narrow line widths that make $^{13}$C-TROSY-HMQC spectra suitable for the complex [17]. None of the residues identified in the functional epitope contain methyl groups, but FimH LD has 64 $CH_3$-containing residues (102 methyl groups total) that are well distributed throughout its structure. NMR assignments are available for 46 of the residues, providing good coverage throughout the LD. Approximately 55 methyl groups have some surface accessibility while others are buried.

LD-Fab824 complex was purified by size exclusion chromatography from a mixture of uniformly $^{13}$C-labeled LD and natural abundance Fab824 in the absence of αmm. $^{13}$C-TROSY-HMQC spectra focused on methyl resonances were collected on LD unbound and bound to Fab824 (black and blue spectra, respectively, in Fig 3A). Most resonances fully overlay between the two spectra, indicating there is no global conformational change upon Fab824

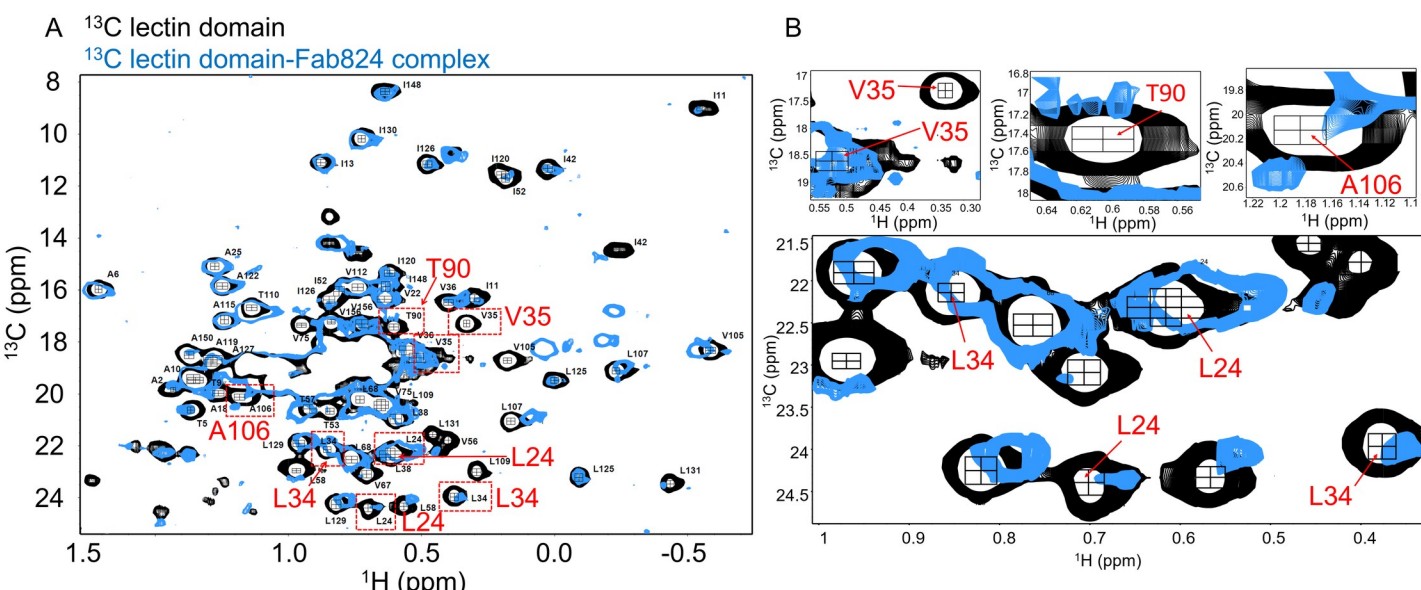

**Fig 3. $^{13}$C-methyl TROSY HMQC spectra showing perturbations in the lectin domain due to Fab824 binding.** The spectrum of free LD is shown in black and the lectin domain in complex with Fab824 is shown in blue. The peaks are not fully filled in to provide clarity. Examples of the largest perturbations are highlighted in red boxes. Note that each methyl group of a sidechain gives its own peak (i.e., two peaks/residue for Ile, Leu, Val). (B) Expanded regions of the NMR spectra showing examples of residues with large perturbations.

binding. This is consistent with the notion that the antibody traps LD in the conformation initially recognized. A subset of resonances, however, exhibits large perturbations indicative of changes in local chemical environment consistent with either a direct contact with antibody or an indirect conformational effect [18]. The types of perturbations observed include: 1) no overlapping (blue) peak and 2) loss of intensity (small blue peak) (Fig 3A and S3 Table). Perturbations were observed for both buried and solvent-exposed methyl groups (expanded spectra of solvent-exposed residues are shown in Fig 3B). We reasoned that large perturbations to buried methyl groups cannot be caused directly through antibody contact and must be affected indirectly (see Discussion) while NMR perturbations of solvent-exposed methyl groups could be due either to direct antibody contact, as would be expected if they are part of the antibody-binding contact surface, or to an indirect effect [18].

Mapping the largely perturbed solvent-exposed residues onto the HAS LD structure (PDB 3zpd) reveals that a majority are either adjacent to the functional epitope (e.g., T90) or form a contiguous patch in proximity to it (L24, L34, V35, and A106) (Fig 2B, green). Surface methyl groups that are near to perturbed residues but whose NMR resonances are not affected by antibody binding are shown as dark blue (Fig 2B). Importantly, there are no unperturbed methyl-containing surface-exposed residues between the NMR-defined patch and the mutagenesis-sensitive patch. Thus, it is reasonable to conclude that the methyl-containing surface-exposed NMR-perturbed residues T90, L24, L34, V35, and A106 are within the contact area of mAb824. The most parsimonious conclusion is that the antibody-contact surface is composed of the functional epitope, the NMR-perturbed surface residues, and the intervening surface, for a total area of 854 $Å^2$ (as measured on PDB 3zpd), i.e. within the expected range of sizes of a structural epitope (Fig 2C).

## L34 and V35 switch between buried and exposed orientations in the HAS and LAS

Comparison of atomic-level structures of LAS and HAS of FimH$^{wt}$ reveals that the NMR-perturbed surface-exposed residues T90, L24, and A106 have similar SASA values in both states (S2 Table). In contrast, L34 and V35 are markedly different in their solvent exposure in the two states. V35 is buried in the LAS and exposed in the HAS (SASA 0 and 70 $Å^2$, respectively) (Fig 2D and 2E). In the LAS, the V35 side chain interacts with side chain atoms of seven other buried amino acids, with a total contact surface of 77 $Å^2$ (S4 Fig). The change in L34 results in its going from almost complete burial in the HAS (1 $Å^2$) to solvent-exposed in the LAS (SASA 26 $Å^2$, partially shielded by side chains of neighboring solvent-exposed residues) (Fig 2D and 2E). Core-buried L34 in the HAS contacts side chains of six residues (total contact surface of 92 $Å^2$), all but one of which are different from those that surround the V35 in its core-buried position (S4 Fig). Taken together, these observations imply that the concomitant switching of L34 and V35 side chains between core and surface positions results in a distinct arrangement of the surrounding core residues.

We performed a targeted molecular dynamics (TMD) simulation to predict conformational changes that may occur during the transition of the LD from the LAS to HAS (see Materials and methods). The chain of events during the TMD simulation is depicted in the S1 Movie file. Concurrent with L34 switching from its surface-exposed orientation in LAS to its core-buried orientation in HAS (Fig 4A), a rupture of hydrogen bonds between backbone atoms of the V35-neighboring V36 on one β-strand and L107 on the adjacent β-strand is predicted (Fig 4A). Such an event is associated with formation of a 6 Å gap between the β-strands. This presumably high-energy "transition" state is followed by the core-to-surface switching of the V35 side chain (Fig 4B) and finally, resealing of the V36-L107 gap (Fig 4B). Thus, the TMD

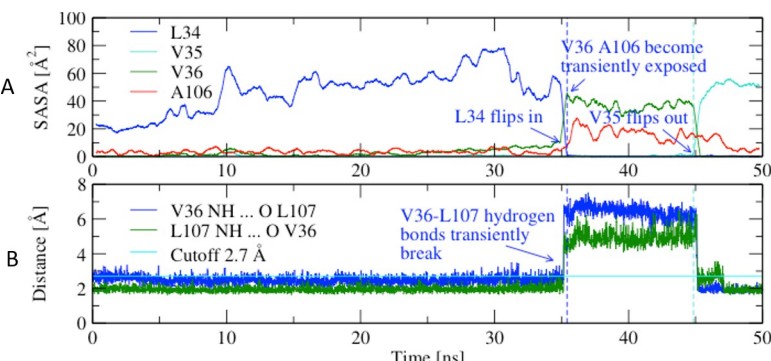

**Fig 4. Conformational changes during a simulation of LAS to HAS transition.** (A) SASA of L34 and V35 side chains. The values are averaged over a 0.5 -ns time window. Vertical dashed lines indicate the time point when L34 flips in (blue) and the time point when V35 flips out (red), respectively. Flipping in or out, respectively, was defined when the SASA of the respective side chain (in the 0.5-ns time averaged values) became less than or larger than 2 $Å^2$, respectively. (B) Distances between atoms involved in the backbone hydrogen bonds between V36 and L107. Horizontal cyan line indicates the distance cutoff used to define a hydrogen bond, i.e., 2.7 Å.

simulation predicts that the switching of L34 and V35 between core and surface involves a high-energy transition state that is substantially different from either the LAS or HAS endpoint structures.

## Alanine replacement of L34 or V35 reduces mAb824's ability to block the allosteric transition

As shown in S1 Table, replacement of L34 or V35 with a smaller hydrophobic alanine (or any other amino acid tested) does not affect mAb824 binding to FimH. We tested the effect of L34A and V35A substitutions on the ability of FimH to adopt LAS and HAS conformations. FimH$^{L34A}$ fimbriae demonstrated FimH$^{wt}$-like ability to bind mAb21 without αmm or under increasing αmm concentrations (Fig 5A), suggesting that substitution of L34 to alanine did not affect the mannose-dependent transition between LAS and HAS. FimH$^{V35A}$ exhibited slightly increased binding of mAb21 in the absence of αmm and required a lower αmm concentration to fully adopt the HAS conformation, suggesting a modest shift in equilibrium away from the LAS and towards the HAS (Fig 5A).

Having established that both alanine mutants retain the ability to transition between conformational states, we investigated whether LAS-HAS transitions in FimH$^{L34A}$ and FimH$^{V35A}$ are restrained by mAb824. Like FimH$^{wt}$, each variant binds to mAb824, but unlike the wild-type protein, the mAb824-bound mutants are still able to shift completely from LAS to HAS in the presence of αmm (Fig 5B and 5C, compare to Fig 1F). Furthermore, mAb824 binding does not stabilize the αmm-induced HAS conformations of FimH$^{V35A}$ or FimH$^{L34A}$ because neither mutant was recognized by mAb21 when αmm is washed out of the mAb824-FimH complex (Fig 5D). Thus, when either L34 or V35 is replaced with a less bulky alanine, the antibody's ability to block the transition between the conformational states is reduced.

## Hydrophilic replacement of V35 or L34 stabilizes HAS or LAS, respectively

Unlike the alanine replacements, substitution of L34 and V35 with a charged glutamic acid resulted in profound changes. FimH$^{V35E}$ is fully bound by mAb21 even in the absence of αmm (Fig 6A), strongly resembling the phenotype of the FimH$^{FocH}$ variant where the HAS conformation is predominant. In contrast, FimH$^{L34E}$ was only marginally recognized by mAb21 even at the highest concentration of αmm tested (Fig 6A). When the L34E LD was fused to the

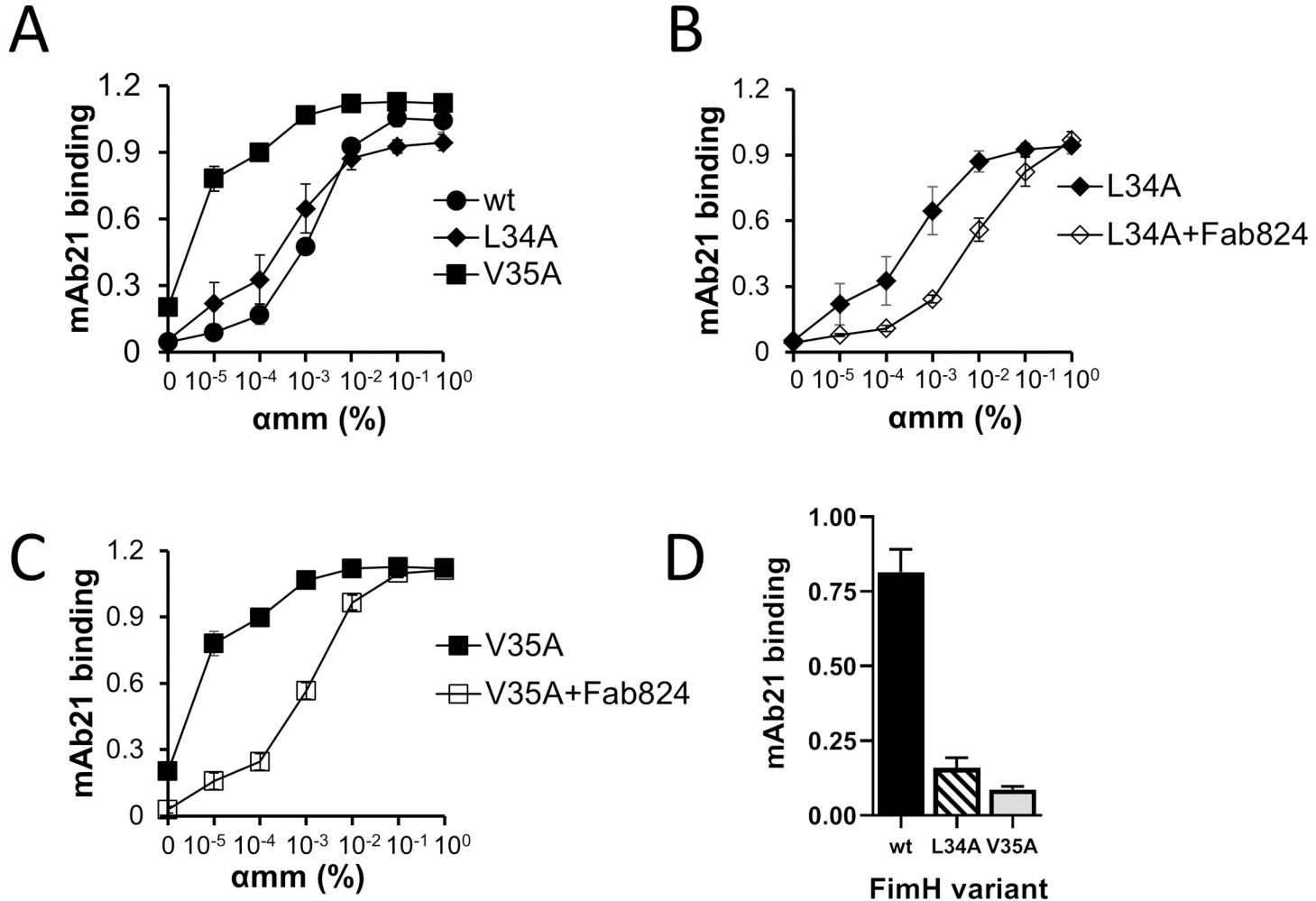

**Fig 5. Effect of V35A and L34A mutations.** Experiments measure HAS-specific antibody, mAb21, binding FimH$^{wt}$ and the two alanine mutants as a function of αmm. (A) mAb21 binding to the three FimH variants. (B) Effect of Fab824 pre-treatment of FimH$^{L34A}$ on mAb21 binding. (C) Effect of Fab824 pre-treatment to FimH$^{V35A}$ on mAb21 binding. (D) Recognition of FimH$^{wt}$ and the mutants by mAb21 following pre-incubation of FimH with Fab824 in the presence of 1% αmm and extensive washing from αmm before mAb21 addition. Data (A-D) are mean absorbance values at 650 nm ± SEM, n = at least 2 independent experiments.

structurally altered PD (i.e., FimH$^{FocH}$), the mAb21 recognition was almost fully restored even in the absence of αmm, indicating that LD$^{L34E}$ assumes a properly folded HAS when the inter-domain interaction is disrupted. Thus, the absence of mAb21 binding to FimH$^{L34E}$ cannot be attributed to misfolding or low expression, implying that the L34E replacement interferes with the LAS to HAS transition.

The effect of glutamic acid mutations just described correlates with the ability of immobilized fimbriae to bind soluble HRP (Fig 6B). In FimH$^{wt}$, V35E bound HRP at least as well as FimH$^{FocH}$, while L34E exhibits marginal binding. In the FimH$^{FocH}$ PD background, the L34E mutant binds HRP strongly, though not as strongly as FimH$^{FocH}$ or FimH$^{V35E}$. As expected, FimH$^{V35A}$ binding to HRP was only slightly enhanced relative to FimH$^{wt}$, while the L34A replacement did not affect the binding to a notable level.

Due to its negative charge and size, burial of a glutamic acid side chain is disfavored, especially relative to valine or leucine. Our results indicate that replacing either L34 or V35 with a Glu stabilizes the LAS and HAS of FimH, respectively, and in a manner that allows the Glu side chain to remain on the surface and avoid burial. Therefore, the orientations of L34 and

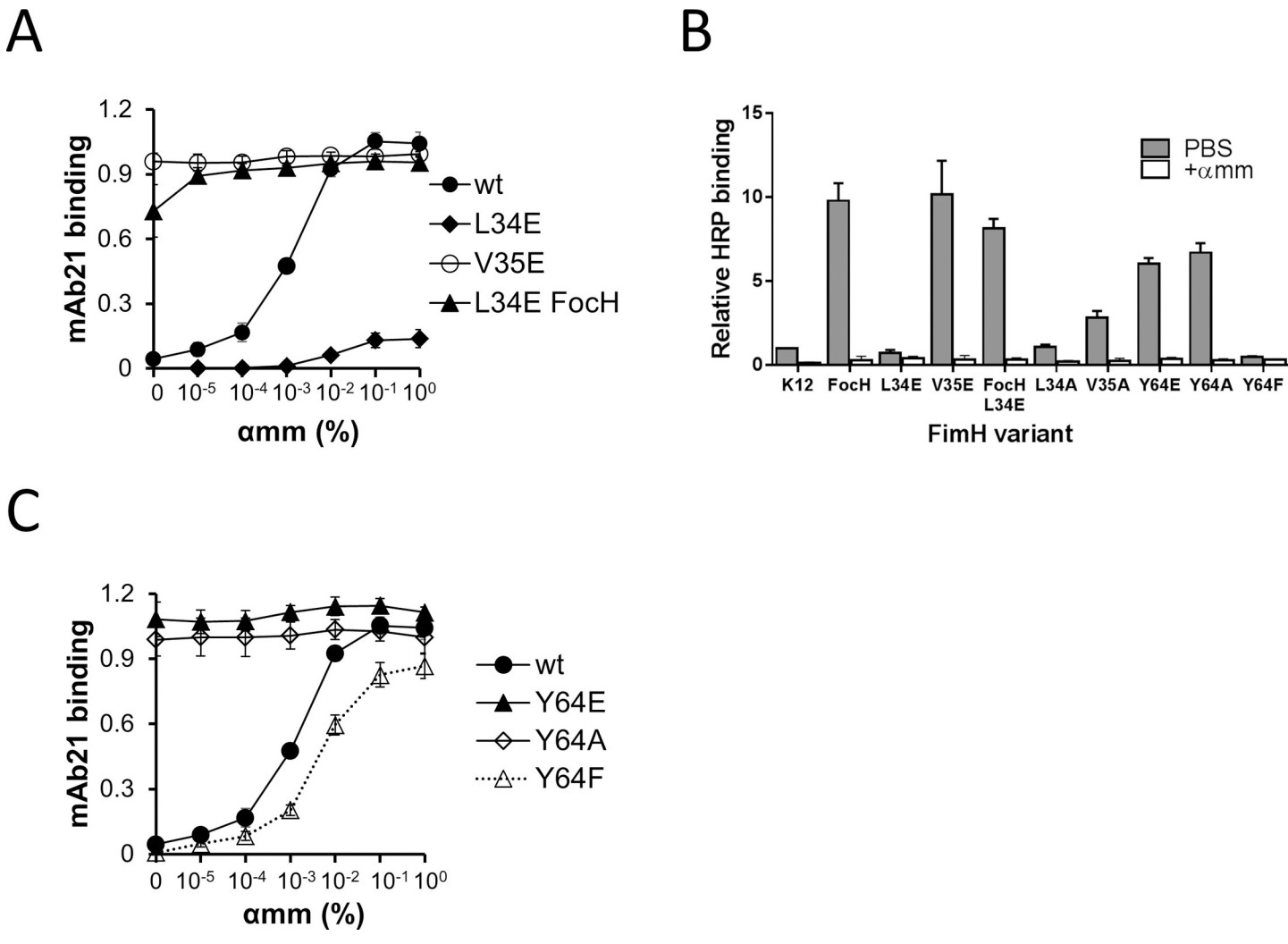

**Fig 6. Effect of mutation of the switch residues on FimH conformational change and function.** (A and C) Presence of the HAS conformation of FimH[wt] and mutants was detected by the HAS-specific mAb21 antibody in ELISA assays. Plate-immobilized fimbriae carrying wild type and mutated variants of FimH were probed with mAb21 in the absence and presence of serially diluted αmm. The data are absorbance values at 650 nm. (B) Relative binding of the mannose-rich glycoprotein (horseradish peroxidase, HRP) to the plate-immobilized fimbriae in the absence and presence of 1% αmm. Data (A-C) are mean ± SEM, n = at least 2 independent experiments.

V35 are critical for maintaining the LD in either the LAS or the HAS. We conclude that these two residues within the mAb824 contact area act as conformational 'toggle switches.'

## L34/V35-like switching is common and occurs in a concerted fashion

Although the total SASA of LD in the LAS (including the interdomain interface) and HAS are nearly identical (7559 and 7460 Å$^2$, respectively), the identity of residues that are core-buried or surface-exposed differs in the two states. Twenty-nine LD residues experience a significant change in SASA (both >2-fold and >20 Å$^2$) between the conformations (Fig 7A). Thirteen residues are more solvent-exposed and 16 residues are less solvent-exposed in the HAS compared to the LAS. Nearly half of these residues (13 out of 29) are aliphatic or aromatic and most are in regions identified to undergo conformational changes, namely, the swing loop, insertion loop, β-bulge and α-switch, as well as the mannose-binding pocket (Fig 7A and 7B)

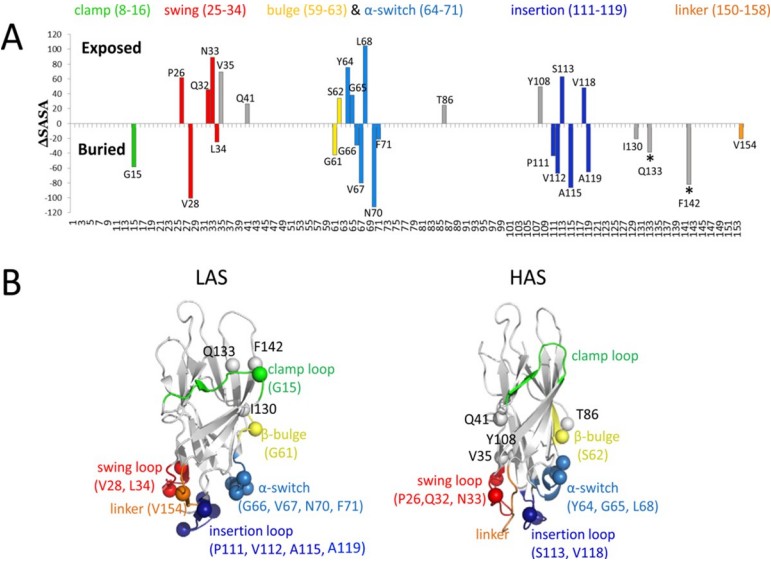

**Fig 7. Residues undergo changes in solvent-accessible surface area (SASA) between LAS and HAS of FimH.** (A) The difference in solvent-exposed surface area (ΔSASA) calculated individually for each residue using LAS and HAS conformers of FimH (PDB code 4xo9 and 4xo8, respectively). FimH residues whose SASA changed by 2-fold and at least 20 Å$^2$ are shown. The regions that exhibit substantial conformational differences in atomic-level structures of FimH LD LAS and HAS [2,3] are color-coded in Panels A and B. Mannose-contacting residues are marked by asterisks. (B) Cα atoms of FimH LD residues highlighted in Panel A are shown as spheres on LAS and HAS structures. Residues that transition from exposed-to-buried (downward-facing bars in Panel A) are mapped on the LAS conformer (PDB code 4xo9) and those that go from buried to exposed (upward-facing bars in Panel A) are mapped on the HAS structure (PDB code 4xo8).

[7,19]. Overall, a significant change in SASA values is observed in about a quarter of residues with low solvent exposure (SASA <20 Å$^2$) in LAS or HAS (S2 Table).

Examination of side chain SASA values for the residues with large differences in the two states along the 5.8-μs long MD simulation from an earlier study [20] revealed that 14 of the residues have a strong association in the timing of their solvent exposure with at least one other residue, with the Pearson's correlation coefficient *R* being either above 0.7 or below -0.7 (S5A Fig). Eleven residue pairs exhibit a positive relationship, i.e. their SASA values co-varied in time in the same direction, while another eleven pairs exhibit a strong negative relationship with the SASA values changing together in time but in opposite directions. Notably, the SASA changes of V35 correlate positively with four residues—Y64, L68, F71, and V118 –more than any other switching residue.

Altogether the observations indicate that, in addition to L34 and V35, a sizable proportion of LD residues undergoes significant changes in orientation between LAS and HAS. The implication is that the FimH conformational transition is associated with a major repacking of side chains within the core of the LD. Moreover, the strong correlation in the timing of changes suggests that those changes occur in a concerted fashion.

## Y64 also functions as a toggle switch

We studied one of residues that closely follows the switching pattern of V35 in more detail. Tyrosine-64 is core-buried in the LAS (SASA 13 Å$^2$) and solvent-exposed in the HAS (SASA 88 Å$^2$) (Fig 2D and 2E). In LAS, Y64 and V35 side chains are in close contact and the timing of their switch out of the core is positively correlated with *R* = 0.75 (S5B Fig). In its solvent-facing orientation in HAS, Y64 is on the opposite face from the putative mAb824 contact area and is 17 Å from the closest residue of the functional epitope.

Similar to the V35E mutation, Y64E adopts the HAS conformation even in the absence of αmm (Fig 6C), indicating that the core orientation of Y64 is important for LAS stability. Substitution of Y64 with alanine also shifts FimH to the HAS (Fig 6C). Replacement of the tyrosine to a structurally similar but more hydrophobic phenylalanine resulted in FimH that requires slightly higher αmm concentrations than FimH^wt to adopt the HAS (Fig 6C). This is consistent with the more hydrophobic phenylalanine ring favoring the LAS (where it will be buried) relative to the HAS (where it will be surface-exposed). The conformational states of the Y64 mutants as detected by mAb21 fully correlate with their HRP-binding capability, with FimH^Y64E and FimH^Y64A exhibiting strong HRP binding and FimH^Y64F showing reduced binding (Fig 6B). Altogether, the results identify Y64 as another SASA-switching residue that functions as a toggle switch critical for the allosteric transition of FimH.

## The allosteric toggle switches are conserved across a broad range of fimbrial adhesins

To explore whether other bacterial adhesins might have FimH-like allosteric switches, we analyzed proteins deposited in the Pfam database [21]. The analysis identified over two dozen representative proteins with structural homology to the FimH LD with a broad range of primary structure identity—from 88% to 17% (S4 Table and S6 Fig). All FimH homologues were designated as known or putative fimbrial adhesins found in various enterobacterial species, including common human pathogens like *E. coli*, *Klebsiella*, *Proteus*, *Citrobacter* and *Serratia*. However, only two proteins in the list have a known ligand specificity–the mannose-binding adhesin, FimH, from *Klebsiella pneumoniae* (accession # A6TDM0, 88% identity to *E. coli* FimH) and galactose-binding adhesin, FmlH, from *E. coli* (accession # P77588, 41% identity to *E. coli* FimH). None of the adhesin proteins has been reported to be allosteric, i.e., able to shift between functional states through action at a distance.

Sequence alignment revealed that positions around L34/V35 and Y64 are highly conserved (S6 Fig). The corresponding amino acid homologues could be identified with certainty in proteins with as low as 34% identity to FimH. Positions homologous to L34 and V35 were identical to FimH or contain a structurally similar isoleucine. The Y64 homologous position is either tyrosine, phenylalanine or, in one case, leucine.

To test whether amino acids homologous to FimH L34, V35, and Y64 might function as allosteric switches, we focused on FmlH, where L32, V33, and F63 are the predicted homologues, respectively (Fig 8A). FmlH is known to bind to terminal galactose (β1–3) N-acetyl-galactosamine, although with low affinity [22]. When full-length wild-type FmlH (FmlH^wt) was expressed in type 1 fimbrial background, no bacterial binding was detected to asialated fetuin, which contains oligosaccharides with galactose (β1–3), even after spinning the bacteria down to the asialofetuin-coated surface (Fig 8B). Remarkably, substitution of glutamic acid for either V33 or F63 yielded mutants with a highly pronounced and galactose-inhibitable binding phenotype (Fig 8B). An L32E mutation did not increase binding. To determine whether L32E inhibits the binding properties of FmlH, its lectin domain was fused to the heterogeneous pilin domain used for the construction of HAS-favoring FimH^FocH variant. Like FimH^FocH, the FmlH^FocH variant containing wild-type FmlH LD exhibited increased binding relative to the wild-type adhesin (Fig 8B). In this context, introduction of the L32E mutation decreased the binding properties of FmlH^FocH in an even more pronounced manner than L34E in FimH^FocH (Fig 6B). Thus, V33E and F63E mutations resulted in activation of the binding phenotype in FmlH, while L32E had the opposite effect, equivalent to phenotypes observed for the corresponding amino acid replacements in homologous positions in FimH.

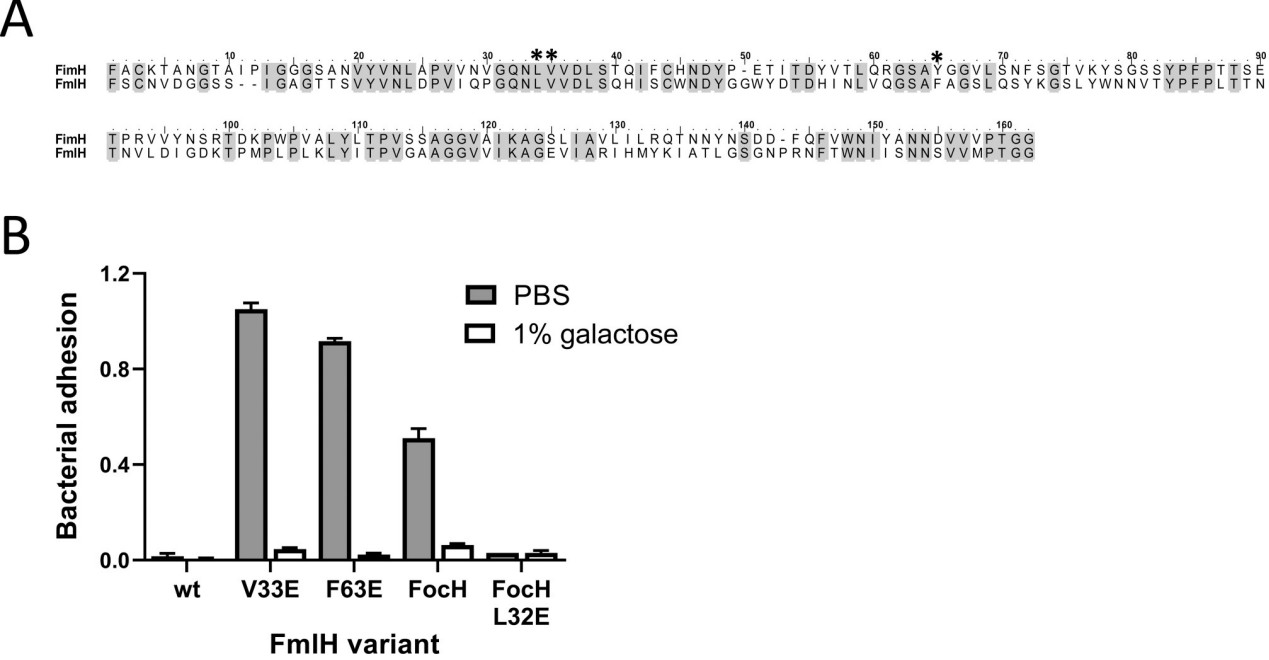

**Fig 8. Toggle switches are conserved in the FimH homologue, FmlH adhesin.** (A) Sequence alignment of FimH and FmlH lectin domains with L34/V35- and Y64-switches marked by asterisks. Identical residues are gray-shadowed. (B) Adhesion mediated by mutant variants of FmlH adhesin in the absence and presence of 1% galactose to surface-coated asialofetuin. Data are mean absorbance values at 600 nm ± SD, n = 1 representative experiment.

## Discussion

Our study shows that 1) a monoclonal antibody that binds both HAS ('active') and LAS ('inactive') conformations of the FimH adhesin blocks the allosteric transition between the states in both directions, 2) restraint of the conformational transition results in elimination of bacterial adhesion under dynamic shear conditions, 3) the antibody-FimH contact area includes toggle-switch-like residues that must transit between core-buried and solvent-exposed orientations during the transitions, and 4) similar allosteric toggle switches are found in other FimH regions and across functionally and structurally diverse fimbrial adhesins of enterobacteria.

In contrast to previously discovered antibodies against allosteric proteins that recognize only one conformational state [23–29], the mAb824 antibody described here binds both and traps the protein in the state to which it initially binds. The binding-critical, functional epitope of mAb824 is structurally similar in both conformations, providing a rationale for how both conformations are recognized by mAb824, but not for how it can trap each state. Antibody-antigen contact areas (i.e., structural epitope) are known to extend well beyond the residues that are energetically critical for binding [14,30,31]. Precise mapping of antibody-antigen interfaces requires atomic-level structures, but these remain challenging to obtain [14]. We used methyl group-specific NMR and identified a region on the LD surface that is close to the functional epitope and exhibits substantial spectral perturbation upon mAb824 binding to FimH. We assign this region as the contact area for mAb824. The proposed antibody-LD contact area does not overlap with the mannose-binding pocket, consistent with the observation that αmm does not inhibit mAb824 binding to FimH even at the highest concentrations tested.

Unlike the functional epitope which is retained in the two conformations, the broader mAb824-LD contact area includes residues whose orientations switch between core-buried

and surface-exposed, namely L34 and V35. Replacement of either of these aliphatic residues with a charged glutamate revealed that the surface-exposed orientations of L34 and V35 are critical to the LAS and HAS states, respectively. It has been shown that there is little free space between antibody and protein antigen atoms in antibody-antigen contact areas, suggesting that longer and/or bulkier side chains of epitope surface residues lie relatively flat [14]. This property implies that antibody binding is likely to restrict the side chains of L34 and V35 on the surface and hinder their switching in and out the core. Such a restriction would in turn block the shift from LAS to HAS and vice versa. In addition, TMD simulations predict that the L34/V35 switching requires transient formation of a large (~6 Å) gap between neighboring β-strands that would represent a substantial perturbation in structure that may also be restrained by mAb824 binding. Importantly, when a smaller alanine sidechain is introduced at either position 34 or 35, mAb824 does not block the transition as effectively. Composed of a single methyl group, surface-core switching of alanine is likely less restrained. The alanine results support our hypothesis that mAb824 binding to FimH inhibits both LAS->HAS and HAS->LAS conformational transitions by providing a steric block to reorientation of residues between their core-buried and surface-exposed orientations. This in turn reveals a molecular mechanism for the inhibition of allosteric transitions by mAb824 that has not been previously described for an antibody-antigen interaction to our knowledge.

Taken together, the results suggest that mAb824 acts as a kinetic trap that blocks the conformational transitions in either direction rather than a thermodynamic sink that stabilizes a specific conformation as in the case of allosteric antibodies. In this new paradigm, mAb824 binding increases the free energy of the transition state between LAS and HAS rather than lowering the free energies of the end-state conformations (S7 Fig). Because the mannose binding and domain separation are reciprocal allosteric events, one facilitating another, the mannose- or tensile force-induced domain separation and, thus, LAS to HAS transition is likely to involve same molecular mechanism of allosteric propagation through the LD, either from the binding pocket to the interdomain interface or in the opposite direction. In turn, the kinetic trapping mechanism is likely to occur under either equilibrium or dynamic conditions, as supported by the functional studies below.

The novel kinetic trapping mechanism provided an opportunity to compare functional properties of the alternate conformations of FimH when they are unable to exchange. We tested the interaction of FimH$^{wt}$ and FimH$^{FocH}$ fimbriae with RNAseB and found that $k_{on}$ is ~50-fold higher and $k_{off}$ is nearly 1000-fold higher for FimH$^{wt}$ relative to FimH$^{FocH}$, resulting in ~10-fold lower affinity. It has previously been estimated that the affinity of LAS towards mannose ligands is ~300–3,000-fold lower than that of HAS [8,9,11,32]. RNAse B's abundant Man$_5$ oligosaccharide moieties are likely the source of the smaller difference in affinity for RNAseB between native FimH and FimH that is trapped in HAS. Among natural ligands, Man$_5$ has the highest affinity to FimH LAS conformers, likely due to the terminal oligomannose in the moiety Man($\alpha$1–3)Man($\beta$1–4)(GlcNAc)$_2$-Asn that interacts with LD via a structurally extended binding site [33]. The larger affinity differences previously reported were based on measurements made with either mannose derivatives or mannosylated oligosaccharides that lack the specific terminal oligomannose component. Those ligands interact with FimH via the mono-mannose mechanism that results in strong binding for FimH in the HAS, but not in the LAS conformation, in contrast to the oligo-mannose (also called 'tri-mannose') binding mechanism [9,10].

Despite the N-linked Man$_5$ oligosaccharide moiety that binds strongly to all natural FimH variants, RNAse B did not bind at detectable levels to the mAb824-trapped LAS conformation of FimH$^{wt}$. As mAb824 does not *directly* block the mannose-binding pocket, its inhibitory effect must be due to disabling the transition from the LAS to HAS. Hence, even when binding

to its strongest natural ligands, FimH must convert from the wide-pocket LAS conformation to a narrow-pocket conformation observed in either the HAS or the intermediate state of FimH. Indeed, ligand binding to the HAS conformation of FimH occurs at a substantially reduced on-rate relative to native FimH [8,32]. While mAb824 does not fully block RNAse B binding to HAS-favoring FimH[FocH], it slowed the on-rate for ligand binding even further. The most parsimonious explanation for these observations is that the ligand-binding pocket visits both closed and open conformations in either state and the frequency with which this occurs determines the on-rate. Thus, while FimH that is trapped in the HAS conformation can bind ligand, it can do so only under conditions that allow ample time for ligand entrance into the binding pocket.

Guinea pig RBC aggregation is the classical test for type 1 fimbriated bacteria binding under dynamic shear conditions. The surface of these cells is rich in mannosylated glycoproteins and aggregation is induced under continuous rocking of the bacterial and RBC suspension. Under those conditions, FimH acts as a catch-bond adhesin: it binds rapidly to cell surface mannose and then strengthens its binding to resist detachment under tensile force. In the 'kinetic selection' mechanism for such binding, the LAS conformation initiates ligand binding and because it has an open binding pocket, the on-rates are very fast [32]. Sustained binding requires transition to the HAS with its narrow pocket and slow dissociation. Tensile force favors this transition by inducing domain separation of the LD and PD and is only applied once FimH is bound to ligand, so this mechanism effectively accomplishes the catch bond function of FimH. mAb824's ability to completely inhibit bacteria expressing FimH[wt] to aggregate RBCs confirms the validity of the kinetic selection model that allosteric conformational shifts are critical for FimH-mannose binding under dynamic conditions. mAb824 represents a novel type of highly effective adhesion-inhibiting antibody that does not act by direct obstruction of the binding pocket but rather by disabling the catch bond properties of FimH.

The mechanism by which L34 and V35 control the conformational, and thus, functional states of FimH through reorientation of their side chains in and out of the protein core is analogous to the action of toggle switches, in which two alternating positions define the on/off status of a device. Alternative conformation-specific orientations of side chains have been reported in other allosteric proteins, e.g. periplasmic (maltose) binding protein MalE [34], rhodopsin [35], integrin [36], DNA-binding receptor NtrC [37], and anti-thrombin III [38]. However, the consequences of the toggle switch-like effect on the conformation and function of allosteric proteins has not been studied thoroughly, nor are they fully appreciated. Our successful activation or de-activation of FimH (and also FmlH adhesin as discussed below) by glutamate mutations at L34/V35 suggests that replacement of hydrophobic residues well removed from ligand-binding pockets with charged residues can serve as a strategy to identify putative toggle switches and discover alternate functional states ("E-scanning" in the case of glutamate substitutions). In support of the E-scanning approach, glutamate replacement of FimH surface-core switching residue Y64 produces a "trapped" HAS conformation, while substitution with phenylalanine, which is slightly more hydrophobic, results in a slightly more stable LAS. These results confirm the importance of specific side-chain orientation for the conformational transition.

The importance of core rearrangements in the FimH allosteric transition is supported by the observation that dozens of residues in the LD switch between primarily core-buried and solvent-exposed orientations in LAS and HAS, nearly half of which are aliphatic or aromatic. The sheer number of residues involved implies that core repacking occurs during the allosteric transition, consistent with the notion that the structural architecture and dynamics of the hydrophobic core play a vital role in the allostery of proteins [39–41]. Given the closely packed nature of protein cores, it is reasonable that repacking occurs in a concerted fashion.

Consistent with this notion, we observe that the timings of the orientational change of many core residues exhibit strong positive or negative correlation with each other in simulations. This suggests that during the allosteric transition in FimH, a change in one residue depends on and causes change in other residue(s) or, alternatively, changes in multiple residues co-depend on a single trigger event. If critical residues are hindered from adopting their new positions during the conformational change, the entire transition may be effectively blocked. We believe that the toggle switch residues represent such positions.

Binding of mAb824 to a surface region of FimH LD that contains toggle switch residues causes perturbations to core-buried residues, as revealed by our NMR analysis. While the nature of these perturbations remains to be determined, many of the perturbed core-buried side chains are in direct contact or proximal to the core-surface switches. The observed NMR spectral broadening is consistent with a change in the internal dynamics upon antibody binding.

Finally, the broad applicability of the allosteric toggle switches and E-scanning concept is supported by our analysis of other fimbrial adhesins. While hundreds of known or putative bacterial adhesins have been characterized and some were shown to mediate catch bond-like adhesion [42,43], none have been shown to be allosteric or to shift between alternate functional states besides *E.coli* FimH. Pfam database analysis suggested that enterobacterial adhesins with as little as 17% sequence identity to FimH are structurally homologous to it. Enterobacteria is the largest and most diverse group of Gram-negative bacterial pathogens causing a variety of infections in humans, with the fimbrial adhesins being among the major virulence factors [44,45]. The L34/V35 and Y64 positions are highly conserved across FimH homologues, suggesting that the allosteric toggle switch function of the homologous amino acids may be preserved. We confirmed this prediction for one of the few FimH homologues with a known binding ligand—the galactose-binding adhesin, FmlH, positioned on the tip of F9 fimbriae that are implicated in the urovirulence of *E. coli* [22]. Glutamate substitution of the three putative toggle switch residues can both activate and inactivate the galactose-specific binding of FmlH. Our results strongly suggest that FmlH has HAS and LAS conformations and, thus, may mediate adhesion via the allosteric catch bond mechanism (currently under investigation). We propose that a broad range of bacterial and non-bacterial adhesion/attachment proteins are allosteric and use similar types of molecular toggle switches to transit between LAS and HAS states. Moreover, the E-scanning technique could be used to induce the HAS, strongly ligand-binding state in allosteric proteins to enable their detailed characterization or even to identify ligands.

In summary, our study provides deeper mechanistic insights into the molecular mechanism and kinetics of allosteric transitions in FimH, and potentially, in a broad range of cell adhesion proteins. We have defined experimental approaches to trap the LAS and HAS conformations of an allosteric protein by a novel type of antibody and to induce their specific functional states by specific types of mutational alterations in the search for allosteric toggle switch residues.

## Materials and methods

### Bacterial strains, fimbriae and reagents

The recombinant *Escherichia coli* K12 strain (AAEC191A) carrying pPKL114 plasmid containing the entire *fim* gene cluster from the *E. coli* strain K12 (but with the inactivated *fimH* gene) and pGB2-24 plasmid caring *fimH*<sup>wt</sup> was described previously [9,10]. The *fmlH* gene was cloned by PCR from *E.coli* CFT073 strain and inserted into pGB2-24 using *Apa*L1/*Sph*1 restriction sites. Isogenic pGB2-24-based plasmids carrying mutated variants of *fimH* and *fmlH* were obtained by site-directed PCR mutagenesis using primers comprising specific mutations [13]. For type 1 fimbriae expression, bacteria were grown overnight in 1l LB

cultures with mild shaking. The type 1 fimbriae were isolated from spun bacterial cultures using a magnesium chloride precipitation method [46]. RNase B (Sigma) was biotinylated using Sulfo-NHS-LC-Biotin (Pierce) according to manufacturer's recommendations followed by FPLC purification using Superdex 75 (GE Healthcare Life Sciences).

### Antibodies and Fabs

mAb824 and mAb21 were expressed and purified from serum-free hybridoma cell cultures grown in SFM media (Gibco) as described elsewhere [13,47]. Briefly, antibodies from culture supernatants were first collected using protein G-agarose (Millipore) followed by size-exclusion chromatography using Superdex 200 (GE Healthcare Life Sciences). Purified mAbs were further digested to Fab fragments using agarose-immobilized ficin (Thermo Fisher Scientific) and separated further by FPLC using Superdex 75 (GE Healthcare Life Sciences).

### Epitope mapping

Binding of mAb824 to a library of purified (isogenic) type 1 fimbriae carrying different mutations in the lectin domain of FimH$^{K12}$ (expressed with wild type-FimH$^{K12}$- and mutated-FimH$^{FocH}$ pilin domain) was screened by ELISA [13,47]. We used an extensive library of FimH mutants compiled in multiple previous studies. In cases when candidate epitope residues were found, more targeted mutagenesis was performed. Fimbriae with FimH$^{FocH}$ adhesin were used as a reference against which binding of the antibody to all lectin domain mutant fimbriae was compared.

### NMR sample preparation, data collection and visualization

$^{15}$N and $^{13}$C isotopic labeling of the lectin domain was carried out by growing *E. coli* cells and inducing protein expression (1M IPTG) in minimal media containing $^{13}$C-glucose and $^{15}$N-ammonium chloride as the carbon and nitrogen sources respectively. The Fab824-lectin domain complex was formed by first immobilizing a 5X excess of lectin domain on a nickel resin via its histidine tag, after which Fab824 was added to the nickel slurry and incubated overnight on a nutator at 4°C. The complex was recovered by washing the nickel resin with 20mM sodium phosphate buffer pH 8, containing 300mM sodium chloride (wash buffer) to remove the free Fab824 and then eluting with the wash buffer containing 500mM imidazole. The elution was concentrated and loaded on a size exclusion column (Superdex 75) to separate the excess lectin domain from the Fab824-lectin domain complex.

NMR Methyl TROSY spectra of the Fab824-lectin domain complex were collected on a Bruker 800 MHz AVANCE spectrometer at 298K. NMR samples were prepared in deuterium oxide with 20mM sodium phosphate (pH 6), 1mM EDTA and 100mM sodium chloride. Data were processed using standard protocols in NMRPipe [48]. NMR data were visualized analyzed in NMRviewJ [49], and the FimH lectin domain assignments were obtained from the Biological Magnetic Resonance Databank (BMRB entry 19256).

### ELISA assays

To test antibody binding to FimH, microtiter plate wells were coated with purified fimbriae at a concentration of 0.1 mg/ml in 0.02 M NaHCO$_3$ buffer for 1 h at 37°C. The wells were washed with PBS and quenched for 20 min with 0.2% (wt/vol) BSA in PBS. The immobilized fimbriae were incubated with serial dilutions of pure mAbs for 1h, and after washing, bound antibodies were detected with a 1:3,000 diluted HRP-conjugated goat anti-mouse antibody (Bio-Rad). The reaction was developed at room temperature using 3,3′,5,5′-tetramethylbenzidine (TMB,

Kirkegaard and Perry Laboratories [KPL]), and absorbance was read at 650 nm. In some experiments, binding of the antibodies was examined in the presence of 1% of αmm. All incubation steps in ELISA assays were performed at 37˚C unless stated otherwise.

To test the effect of αmm or mAb824 on mAb21 binding to fimbrial FimH, immobilized fimbriae were incubated for 45 min with 1% αmm, or 25 μg/ml pure Fab824, or 1% αmm and the Fab together. After washing, 0.2 μg/ml mAb21 was added to wells for 45 min. Bound antibodies were detected with a 1:5,000 diluted HRP-conjugated goat anti-mouse Fc antibody (Sigma-Aldrich). In some experiments, mAb21 at a concentration of 0.5 μg/ml was added to fimbriae-coated wells in the absence or presence of serial dilutions of αmm. For simultaneous mAb824/mAb21 binding test, the fimbrial FimH$^{wt}$ was first complexed with mAb21 antibody in the presence of 1% αmm and after wash with PBS probed with biotinylated mAb824 (used in concentration 0.1 μg/ml). Binding of the latter antibody was detected using 1:5,000 diluted HRP-conjugated streptavidin (Invitrogen).

## Red blood cell (RBC) aggregation assay

Guinea pig RBC (Colorado Serum Company) were washed with PBS three times. 50μl of RBC suspension (1%) was mixed 1:1 with 50μl of $4 \times 10^8$ of type 1 fimbriated bacteria in a glass plate in the absence and presence of 1% αmm and continuously mixed at high speed on rotator (Model 260300F, Fisher Scientific) for extended period of time. The aggregation of RBC was visually scored and the time at which noticeable aggregation was observed and reported in minutes (min). In some experiments, bacterial cells were first pre-incubated for 5 min with 67 μg/ml of mAbs in the glass plate and then mixed with RBC suspension in the absence and presence of 1% αmm.

## Horseradish Peroxidase (HRP)-binding assay

96-well microtiter plates were coated overnight with purified type 1 fimbriae in 0.02 M NaHCO$_3$ buffer at pH 9.6 (0.1 mg/ml). The immobilized fimbriae, after blocking with 0.2% BSA in PBS were incubated with horseradish peroxidase (HRP) at a concentration of 50 μg/ml for 1h in the absence or presence of 1% αmm. After extensive washing with PBS, bound HRP was detected by adding 100 μl of TMB substrate (Kirkegaard and Perry Laboratories, KPL) and absorbance was read at 650 nm.

## Bacterial binding assay

FmlH-dependent bacterial adhesion was analyzed as described previously with minor modifications [47]. Briefly, Immulon 4HBX microtiter plates (Thermo Electron Corp.) were coated with 20 μg/mL of asialofetuin (Sigma-Aldrich) and plates were quenched with 0.2% BSA in PBS. Bacterial suspensions in PBS (OD540 = 2) were added to each well in a volume of 100 μl and plates were spun for 2 min at 750 x g. After extensive washing, bound bacteria were stained with 0.1% (vol/vol) crystal violet for 20 min at room temperature and were washed several times with water. Then 100 μl of 50% (vol/vol) ethanol was added to each well, and the absorbance at a wavelength of 600 nm was measured. For inhibition, bacterial binding was tested in the presence of 1% galactose (Sigma).

## Biolayer interferometry

The kinetics of antibody binding to fimbrial forms of FimH was determined by using monoclonal antibodies (mAb) captured on anti-mouse IgG biosensors (ForteBio). In brief, hydrated biosensors (FortéBio) were immobilized with purified mAb824 (applied at a concentration of 5 μg/ml in HEPES buffered saline (HBS, pH 7.4)) supplemented with 0.2% BSA until the signal

reached ~0.5 nm. The antibody-immobilized biosensors were washed with 0.2% BSA HBS and allowed to interact with serially diluted purified fimbriae (25–100 nM) in the absence and presence of 52 mM αmm. After the association phase, the dissociation of mAb-fimbriae complexes was performed by moving the biosensors into 0.2% BSA HBS buffer. mAb antibody-captured biosensors in 0.2% BSA HBS buffer (in the absence and presence 52 mM αmm, respectively) were used for single reference subtraction. The collected data were processed using Data Analysis 7.0 software (FortéBio). The data were globally fitted using a 1:1 Langmuir binding model in GraphPad Prism 6 software. Of note, preincubation of fimbriae with αmm (190 g/mol) did not affect sensorgrams recorded for fimbriae binding with the average molecular weight of fimbriae being 15000 kDa.

To test the kinetics of fimbriae binding to RNase B, streptavidin-coated biosensors (SA, FortéBio) were first allowed to bind biotinylated RNase B (applied at a concentration of 10–20 µg/ml in 0.2% BSA HBS). After the baseline step with 0.2% BSA HBS, the RNase B-captured biosensors were exposed to purified fimbriae (50 and 100 nM concentrations) in the absence or presence of 4µM Fab824 (i.e. in the presence of a 80- and a 40-fold excess of the Fab in relation to purified fimbriae, respectively), or in some experiments 104 mM αmm. The dissociation of the fimbriae-RNase B complexes was conducted in 0.2% BSA HBS. The binding sensorgrams after single reference subtraction (biosensors with 0.2% BSA HBS, or Fab824, respectively) were used for global fitting. Because the response for FimH$^{FocH}$ could be fit with a 1:1 Langmuir binding model, this model was employed for that data Prism 6 (GraphPad) software. Because the 1:1 model could not fit the FimH$^{wt}$ data, the more complex conformational change model, used previously to model native, catch-bond forming FimH variants [32], was employed for that data using BIA evaluation (BIAcore). This is consistent with the notion that HAS is heavily predominant in FimH$^{FocH}$, while FimH$^{wt}$ is in equilibrium between HAS and LAS conformations. The standard error that was given in the table for the kinetic parameters relates to the "fit" of the data to the model.

## Protein structure analysis and visualization

Protein Data Bank entries 4xo9 (chain A) and 4xo8 (chain A) (8) for crystal structures of FimH served as models of the LAS and HAS conformers. In addition, the NMR structure of isolated lectin domain (PDB entry 3zpd (chain A) [50]) was used to present the NMR-determined structural epitope of mAb824. Superposition of appropriate structures were carried out using the SUPERPOSE module [51] in the CCP4 program suite [52], and RMSDs and angle calculations were carried out using locally-written programs. Solvent-accessible surface area (SASA) of different forms of FimH lectin domains was calculated using the areaimol program within the CCP4 program suite [52]. The SASA of functional and extended epitopes was calculated using PyMOL (DeLano Scientific LLC). The spatial distribution of amino acid residues involved in mAb epitopes and distances between Cα carbons were measured using PyMOL (DeLano Scientific LLC). The contact surface area between core-oriented side chains of V35 (in LAS) and L34 (in HAS) and their neighboring (in 3 Å distance) residues was determined using CHARMM [53]. The contact surface area is defined by the difference between surface measured for all contacting side chains (defined in S3 Fig) in the absence and the presence of the core-oriented side chain of V35 (or L34) [54,55].

## TMD simulation

The TMD simulation was started with the LD in LAS derived from the crystallographic structure with PDB code 3jwn [7]. For reasons of efficiency, the pilin domain was removed and the lectin domain was truncated after residue 160 before starting the simulation. During the TMD run the crystallographic structure with PDB code 1uwf [56] was used to steer the protein

conformation towards HAS. The simulation setup was similar as in a previous study [20]. Briefly, the simulation was performed with the program NAMD [57] and the CHARMM22 force field [58]. The protein was solvated using a cubic water box with side length of 86 Å and periodic boundary conditions. Chloride and sodium ions were added to neutralize the system and approximate a salt concentration of 150 mM. During the simulations, the temperature was kept constant at 300 K by using the Langevin thermostat [59] with a damping coefficient of 1 ps$^{-1}$, while the pressure was held constant at 1 atm by applying a pressure piston [60]. Before starting the TMD run, a simulation was performed with LD in LAS for 10 ns without the addition of any forces. After this equilibration phase, forces were applied to the Cα atoms of regions of LD that were identified to significantly change conformation from LAS to HAS in a previous study [20], i.e., residues 22–26, 28–37, 58–69, 71–77, 109–124 and 150–157, and to the heavy atoms of residues that flip between solvent-exposed and core-buried orientations from LAS to HAS, i.e., residues 28, 33, 34, 35, 62, 63, 64, 67, 68, 118 and 119. Forces on these atoms were applied according to their RMSD from the target conformation (LD in HAS) following a TMD protocol [61]. The TMD simulation was performed for 50 ns.

Time series of the side chain SASA values of switching residues were calculated using the program CHARMM [53] along a previously published MD simulation [20]. This analysis included 25 non-glycine residues out of 29 total switching residues (defined in the study, see S5 Fig). The correlation coefficient (R) between the time series values of the SASA (after averaging over a 240-ns time window) was calculated by performing a linear regression with the program xmgrace (https://plasma-gate.weizmann.ac.il/Grace/).

### Structural homology analysis

For the structural homologues search, the conserved protein families' database Pfam (https://pfam.xfam.org/) was screened using the full-length protein sequence of FimH$^{wt}$. Resulting multiple sequence alignments retrieved from Pfam separately for lectin domain were examined for conservation of the allosteric switches (L34/V35 and Y64).

### Supporting information

**S1 Fig. Analysis of mAb824 binding to fimbrial form of FimH$^{wt}$ and FimH$^{FocH}$.** (A) Dose-dependent mAb824 binding in the absence and the presence of 1% αmm to surface-immobilized fimbriae bearing different variant of FimH by ELISA. Data are mean ± SEM, n = 3. (B) Binding of mAb824 (biotinylated) to PBS-treated- and mAb21-complexed FimH$^{wt}$ fimbriae. (C) Binding of FimH$^{wt}$ fimbriae (at 25–100 nM concentrations) to probe-immobilized mAb824 in the absence (green) and presence (orange) of 1% αmm measured by biolayer interferometry (BLI). Kinetic data were globally fitted (black lines) to a 1:1 Langmuir binding model (Prism 6, GraphPad) and obtained kinetic parameters are reported in the text. Data from one representative experiment are shown.
(PDF)

**S2 Fig. Effect of mAb824 on FimH binding kinetics.** Kinetics of FimH$^{wt}$ (A) and FimH$^{FocH}$ (B) fimbriae binding to ribonuclease B (RNaseB) measured by biolayer interferometry. Recorded sensorgrams show binding of isolated fimbriae to RNaseB (biotinylated) captured on a streptavidin-coated probe. Binding of fimbriae (at 50 nM and 100 nM concentrations) in the absence (green), and the presence of Fab824 (blue) or αmm (yellow). The binding data were globally fitted (black lines) to two-state conformation change (FimH$^{wt}$) [32] (A) and a 1:1 Langmuir (FimH$^{FocH}$) (B) binding models.
(PDF)

**S3 Fig. Red blood cell (RBC) aggregation by FimH-expressing bacteria under rocking conditions; the 10 minutes time point.** (**A**) RBC aggregation without pre-incubation with mAb824, with serial dilutions of bacteria (1:1 corresponds to the OD = 1.0); (**B**) RBC aggregation upon pre-incubation with mAb824; at the highest bacterial dose.
(PDF)

**S4 Fig. Solvent-core dynamics of L34/V35 correlates with a solvent-core re-orientation of other core residues.** Altered network of residues in 3 Å distance (green and yellow) of core-buried V35 (left panel) and L34 (right panel) in LAS and HAS, respectively, are shown. The V35-interacting residues are: V36, Y64, L68, L107, L109, I126, V156, and L34-interacting residues are: L24, A25, V28, L109, I120, V154.
(PDF)

**S5 Fig. Correlation between SASA side chains of the switching residues along the 5.8-μs long MD simulation.** (A) A previously published MD simulation [20] performed at 330 K on the Anton supercomputer, a resource dedicated to the production of long MD trajectories, was re-analyzed to study correlations between time series of SASA values of side chains that are observed to change between surface and core orientations when comparing X-ray structures of LAS and HAS. A total of 25 side chains were compared pairwise and the Pearson's linear correlation coefficient was calculated between the 240-ns time averages of their SASA values using the program xmgrace. Color scale indicates the strength of the correlation. (B) Time course of the solvent accessibility of V35 and Y64 along the 5.8-μs long MD simulation [20]. Plotted are running averages over a time window of 240 ns.
(PDF)

**S6 Fig. Sequence alignment of 29 representative structural homologues of *E. coli* FimH lectin domain retrieved from the Pfam database.** The identity/similarity of sequences is marked by red/grey shading. Sequence numbering is the same as in S4 Table. L34, V35 and Y64 positions in FimH^wt (sequence #1) are marked by asterisks.
(PDF)

**S7 Fig. Free energy diagram for FimH conformational changes in the absence and presence of antibodies.** In the absence of mannose, LAS predominates while HAS predominates in the presence of mannose. mAb21 binding further stabilizes HAS. mAb824 traps both HAS and LAS by increasing the transition-state free energy ($\Delta G^{\ddagger}$) required to transit between the two states, lowering the frequency of the switch in both directions.
(PDF)

**S1 Table. The functional epitope of mAb824 as mapped by ELISA using FimH mutant library.**
(PDF)

**S2 Table. Solvent accessible surface area (SASA) of individual amino acid resides in LAS to HAS states of FimHwt.**
(PDF)

**S3 Table. The effect of mAb824 binding on methyl-containing residues of FimH^wt lectin domain as determined by ^{13}C methyl-TROSY HMQC.**
(PDF)

**S4 Table. Pfam database-retrieved structural homologues of *E. coli* FimH adhesin.**
(PDF)

**S1 Movie. Movie showing the TMD trajectory between 34 ns and 46 ns.** During this simulation time frame, the flipping of residues L34 (at 4 seconds in the movie) and V35 (at 37 seconds) was observed. L34 and V35 are colored in blue and red, respectively, and their side chains are shown in the stick and ball representation and labeled. Distances between atoms involved in backbone hydrogen bonds between V36 and L107 are indicated by blue dashed lines. Atoms involved in the V36 NH. . . O L107 hydrogen bond are colored in black while those in the L107 NH. . . O V36 hydrogen bond are colored in green, respectively. (PDF)

## Acknowledgments

The TMD simulation was performed on the Comet supercomputer at the San Diego Supercomputing Center thanks to the XSEDE allocation TG-MCB140143. We thank Brendan Mumey and Thiruvarangan Ramaraj (Montana State University, Bozeman) for helpful discussion.

## Author Contributions

**Conceptualization:** Dagmara I. Kisiela, Pearl Magala, Gianluca Interlandi, Benjamin Basanta, Wendy E. Thomas, Ronald E. Stenkamp, Rachel E. Klevit, Evgeni V. Sokurenko.

**Data curation:** Dagmara I. Kisiela, Pearl Magala, Laura A. Carlucci.

**Formal analysis:** Pearl Magala, Gianluca Interlandi, Laura A. Carlucci, Angelo Ramos, Veronika Tchesnokova, Benjamin Basanta, Vladimir Yarov-Yarovoy, Hovhannes Avagyan, Anahit Hovhannisyan, Wendy E. Thomas, Ronald E. Stenkamp, Rachel E. Klevit, Evgeni V. Sokurenko.

**Funding acquisition:** Laura A. Carlucci, Wendy E. Thomas, Rachel E. Klevit, Evgeni V. Sokurenko.

**Investigation:** Dagmara I. Kisiela, Pearl Magala, Gianluca Interlandi, Laura A. Carlucci, Angelo Ramos, Veronika Tchesnokova, Hovhannes Avagyan, Anahit Hovhannisyan, Wendy E. Thomas, Ronald E. Stenkamp, Rachel E. Klevit.

**Methodology:** Dagmara I. Kisiela, Pearl Magala, Gianluca Interlandi, Laura A. Carlucci, Wendy E. Thomas, Ronald E. Stenkamp, Rachel E. Klevit, Evgeni V. Sokurenko.

**Project administration:** Rachel E. Klevit, Evgeni V. Sokurenko.

**Supervision:** Rachel E. Klevit, Evgeni V. Sokurenko.

**Validation:** Angelo Ramos, Hovhannes Avagyan, Anahit Hovhannisyan.

**Visualization:** Rachel E. Klevit, Evgeni V. Sokurenko.

**Writing – original draft:** Dagmara I. Kisiela, Pearl Magala, Rachel E. Klevit, Evgeni V. Sokurenko.

**Writing – review & editing:** Dagmara I. Kisiela, Pearl Magala, Gianluca Interlandi, Laura A. Carlucci, Wendy E. Thomas, Ronald E. Stenkamp, Rachel E. Klevit, Evgeni V. Sokurenko.

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
