## [Decision Letter · Decision Letter 0]

11 Jan 2021

Dear Dr. Sokurenko,

Thank you very much for submitting your manuscript "Toggle switch residues control allosteric transitions in bacterial adhesins by participating in a concerted repacking of the protein core." for consideration at PLOS Pathogens. As with all papers reviewed by the journal, your manuscript was reviewed by members of the editorial board and by three independent reviewers. In light of the reviews (below this email), we would like to invite the resubmission of a significantly-revised version that takes into account the reviewers' comments.

As you will see from their comments, all reviewers appreciate the interest of your study and its contribution to the field of bacterial adhesion. However, they also raise a number of issues and provide specific comments and suggestions, aiming to improve the quality of the arguments and data presentation.

Reviewer #2 also raises some major issues and questions that may require additional experiments. Please address these questions thoroughly, in particular with regard to experimental conditions and data interpretation in their first point, as well as the differences in affinities between the two states observed here compared to the literature in the second point. Bear in mind however that, in the editorial view, the additional experiments are not essential to meet the criteria for publication in PLOS Pathogens.

Please pay particular attention to describing the methodology in sufficient detail so as to make your data reproducible. Reviewer #1 raises a number of points regarding the figures and suggests a representation of the open and closed states similar to the file attached. I also agree with this reviewer that providing a movie of MD simulations would provide strong visual support and should be included s a supplementary file.

Please make sure that you address the reviewers comments in full in order to avoid multiple rounds of revision.

We cannot make any decision about publication until we have seen the revised manuscript and your response to the reviewers' comments. Your revised manuscript is also likely to be sent to reviewers for further evaluation.

Sincerely,

Olivera Francetic, PhD

Guest Editor

PLOS Pathogens

Guy Tran Van Nhieu

Section Editor

PLOS Pathogens

Kasturi Haldar

Editor-in-Chief

PLOS Pathogens

orcid.org/0000-0001-5065-158X

Michael Malim

Editor-in-Chief

PLOS Pathogens

orcid.org/0000-0002-7699-2064

Reviewer's Responses to Questions

**Part I - Summary**

Reviewer #1: The manuscript "Toggle switch residues control allosteric transitions in bacterial adhesins by participating in a concerted repacking of the protein core" shows an allosteric change in the tip protein of type 1 bacterial adhesion pili upon binding of the lectin domain to its ligand. This conformational change produces a transition from an inactive state with low mannose affinity to an active state with high mannose affinity.

The mechanism of this transition by changes in the lectin domain is characterized in detail in this manuscript, with primary data from an antibody that blocks the transition between states. Additional data supporting the proposed mechanism are included from NMR, mutation studies, adhesion assays, and molecular dynamics. Together, the data and analyses fully support their conclusions.

While it was previously shown by some of the authors of this paper, that binding of these pili is by a catch-bond mechanism, this new work is a great advance forward as it describes the molecular mechanism used to produce the strong bacterial adhesion observed for E coli expressing type 1 pili.

Reviewer #2: This article by Kisiela et al., describes a novel monoclonal antibody (mAb824) that binds to lectin domain of FimH. This antibody recognizes both the low-affinity (LAS) and high-affinity states (HAS) of FimH. The authors conduct binding experiments with wild type FimH and FimH-FocH (represents FimH in a HAS). Another antibody (mAb21) is used in binding experiments to indicate that FimH has adopted the HAS. They show that FimH bound to mAb824 does not transition from LAS to HAS (or vice versa) during mannose titration experiments and suggest the antibody functions as a kinetic trap preventing transitions between FimH states. The authors use a library of FimH mutants to map the mAb824 epitope and use 13C-methyl NMR analysis to map the FimH residues perturbed upon mAb824 binding. L34 and V35 are two residues identified by the NMR experiments that show different levels of surface exposure in the LAS vs HAS of FimH. The authors propose that these residues, amongst others, act as toggle switches that control the allosteric transition of FimH from LAS to HAS and back. Lastly, the authors identify FmlH from E. coli as another FimH-related adhesin with similar allosteric properties. While I think that this manuscript is well-written and will be of interest to the field, I have a number of questions regarding some of the results as well as the terminology used.

Reviewer #3: FimH is located at the tip of E. coli pili and is involved in pathogenesis. FimH is a mannose-specific adhesin of E. coli that has the ability to switch between high/low affinity states. While the high/low affinity states have been structurally characterized, the molecular mechanism of switching is not completely understood.

The authors perform a series of experiments to elucidate the mechanism of switching between high/low affinity states of FimH. They use a monoclonal antibody mAb824 as a probe and show that the antibody likely binds an allosteric site on the lectin domain of FimH (Figure 1). The authors use a comprehensive mutagenesis in order to map the mAb824 epitope (Figure 2). Furthermore, they use the NMR analysis of the FimH-Fab complex to identify residues affected by mAb824 binding (Figure 3). Intriguingly, the authors find several residues – referred to as ‘toggle switches’ – that appear to block high/low affinity switching by preventing re-packing of a number of semi-buried residues. The conclusions are further supported by MD simulations and functional characterization of the mutants of ‘toggle switch’ aa residues (Figure 4 and 5). In addition, the authors suggest that the residues involved in allosteric regulation of adhesin function are conserved in the family of related bacterial adhesins, implicating a common mechanism.

**Part II – Major Issues: Key Experiments Required for Acceptance**

Reviewer #1: No additional scientific experiments are needed. However there are figures that are confusing and/or do not show what the authors are hoping to convey. These are discussed below.

Reviewer #2: 1. The authors have made no attempt to explain why their binding data disagree with other molecular explanations of the FimH catch-bond mechanism. It is my understanding that it was thought that ligand (mannose) binding does not lead to the separation of the pilin and lectin domains and therefore is not directly linked to the transition between LAS to HAS, and that this transition only occurs in the presence of tensile forces. This framework of the FimH catch-bond rationalized how bacteria can further disseminate during infection because the initial low-affinity interaction with mannosylated receptors could be overcome by flagellar motility under low shear conditions. Equally, it provided an explanation of why the interaction of FimH with soluble uromodulin might be less productive than an interaction with cell bound receptors. My question is simply how do the authors explain that mannose titration leads to domain separation and interaction with mAb21? Is there perhaps an alternative explanation for these results? Or were these experiments conducted under shaking conditions? If not, how does this fit in with existing data? Experiment suggestion: could you use NMR to directly show that mannose binding leads to domain separation? To validate that mAb21 binding actually reports on transition to the high-affinity domain separated state.

2. Question: Why have the authors chosen to use the FimHFocH construct as the proxy for the HAS? Would it make sense to also include lectin domain only constructs as a further control in the binding experiments? I was struck by the discrepancy in the reported KD differences between wt and FimH-FocH (~10 fold) compared to the affinity differences previously reported (>1000 fold) for the low and high affinity states using different constructs.

3. I recommend revising the terminology of “active” and “inactive” FimH conformations used throughout the manuscript. In my view, the low and high-affinity states of FimH are not the same as inactive and active FimH.

Reviewer #3: (No Response)

**Part III – Minor Issues: Editorial and Data Presentation Modifications**

Reviewer #1: The pilus type under study is not stated until line 410. It is important to say earlier that FimH is the adhesin from type 1 pili expressed on E coli, and more specifically, on UPEC.

The NMR methodology is not described in a manner that would make the data reproducible. In particular, it would be helpful to include information on the 13C-methyl spectral analysis, or at least provide a reference to how these data are processed and analyzed.

With respect to figures, the introduction (starting at Line 78) would be improved by a figure. Perhaps simply a reference to Figure 1A would suffice. Looking at pdb's 3jwn, 1uwf and 6gtw, it is clear that the shape of the binding site goes from wide open to an almost fully enclosed cavity (see figure). this is not conveyed in the diagram, and it would be helpful if it could be somehow included. Also, I do not see a new large linker appearing between the subunits. Yes, the binding site for mAb21 becomes exposed, but the diagram seems to create an incorrect impression of the change.

Figure 2D does not convey what the authors describe. It is not possible to see what is buried and what is exposed, nor understand what happens to L34 and V35 in the two states.

Figure 3 is not clear. For example, there are 2 labels for each of the residues L 34 , V35 and V36, L107. perhaps color all the labels to match the color of the signal (blue vs black)?

Figure 4 does not show all the data needed to assess and understand the results. There should be a movie showing that at the specified time points L34 flips in and V35 flips out. This need to include movies is true to support all the MD data.

line 270. half turn rotation of the side chain. This is not clear in the figure 2D. What is the rotation axis?

line 284. how does fig 4A show beta strand unfolding before the L34 switch?

line 345. 13 residues are more solvent exposed and 16 are less exposed in the high affinity state. This goes against what is depicted in Fig 1A, so that figure is misleading, with its appearance of a new extended linker that would be exposed.

Fig S4B label should be A119, not A1119

Reviewer #2: Introduction

Line 78: Is it really the case that FimH is the most prevalent adhesin of E. coli? This is a very strong statement and I am not sure this is the case. Especially since chaperone-usher pilus expression (e.g. type 1 pili – FimH, P pili – PapG) is subject to complex regulation over the course of an infection.

Line 83: low affinity state does not equal inactive FimH (it still binds and this low-affinity binding is functionally relevant)

Line 93: typo – “catch-bond interactions”

Results

Line 183: The off-rate is significantly reduced compared to what? This is not clear. There are no statistics provided, so the word “significant” should be avoided altogether.

Lines 244-246: What is the difference between “only a very small blue peak remaining in original position” and “loss of intensity”? Perhaps providing some example residues that exhibit these types of effects on the resonances in Fig 3 would be helpful for the reader.

Lines 257-258: Is this really the case? Does Fig 2 not indicate a gap between patches?

Line 262: There appear to be many more NMR-perturbed residues based on Fig 3? Which criteria were applied to choose NMR-perturbed residues? Please specify in the methods section.

Line 322: Words missing? And “not due” to a misfolded form. What control experiments were conducted to rule this out?

Discussion

Please discuss how these results fit in with the other models of the catch-bond e.g. where mannose binding does not yet lead to full domain separation (perhaps begins to shift the equilibrium), and that the high-affinity state only becomes prevalent under tensile force conditions that separate the PD and LD.

Table 1

• There are two identical kconf rows for both FimH wt and FimH-FocH? Copy-paste error?

• Please explain why you used two different equations for the KD depending on the FimH construct and define what kconf and k-conf are – perhaps in the Methods section.

• Were the RBC agglutination results dose dependent? Such a result would be much more convincing if one could see raw data and not just the time upon which noticeable aggregation was observed.

Figure 2

• What is the explanation for why the mutagenesis-identified patch is not the same as the patch identified by NMR? Or at least overlapping?

Figure 3

There seem to be many more residues perturbed by mAb824 binding than mentioned in the text?

Figure 5

The y-axis scale is quite different for these experiments compared to those in Fig 1

Methods

Is it sensogram or sensorgram?

Reviewer #3: Specific comments:

Line 119: ‘αmm, hereinafter called mannose’ This is a bit confusing, since ‘amm’ is used in the text.

Line: 201: It would be helpful if the rationale of the selection of residues for mutagenesis is explained.

Line 577: ‘Gram-negative bacterial…’

Figure 2, panels B and C. The arrows showing the rotation between the views should point in the opposite direction.

Figure S4. This Figure contains an informative illustration of the structural transition and perhaps could be incorporated in the main text (just a suggestion).

PLOS authors have the option to publish the peer review history of their article (what does this mean?). If published, this will include your full peer review and any attached files.

Reviewer #1: **Yes: **Esther Bullitt

Reviewer #2: No

Reviewer #3: **Yes: **Konstantin V. Korotkov
---

## [Decision Letter · Decision Letter 1]

2 Mar 2021

Dear Dr. Sokurenko,

We are pleased to inform you that your manuscript 'Toggle switch residues control allosteric transitions in bacterial adhesins by participating in a concerted repacking of the protein core.' has been provisionally accepted for publication in PLOS Pathogens.

Best regards,

Olivera Francetic, PhD

Guest Editor

PLOS Pathogens

Guy Tran Van Nhieu

Section Editor

PLOS Pathogens

Kasturi Haldar

Editor-in-Chief

PLOS Pathogens

orcid.org/0000-0001-5065-158X

Michael Malim

Editor-in-Chief

PLOS Pathogens

orcid.org/0000-0002-7699-2064

Dear Dr Sokurenko,

Thank you for submitting the revised version of your article "Toggle switch residues control allosteric transitions in bacterial adhesins by participating in a concerted repacking of the protein core". As the first decision was a Major revision, your article has now been reexamined by the same external experts. It is my pleasure to inform you that the three reviewers agreed that the revised manuscript is much improved and all recommend its acceptance. Please not however, as per request of Reviwer 1, that minor modifications should be made in the molecular dynamics video, by labelling the key residues to improve visualisation of the change, as well and by providing the legend for this supplementary video file.

I would also like to request a minor change in Figure 2, panels B and C, where the direction of the arrows indicating protein rotation is not logical. In the current configuration, a 90° rotation of the molecules shown on the right of each panel in the direction of the arrow would result in a view with the red patch pointing away from the observer. To correct this, please invert positions of the left and right views in each panel and also please reverse the direction of the arrow.

Thank you again for choosing PLoS Pathogens for the publication of your work.

Kind regards,

Olivera Francetic

Guest Editor

Reviewer Comments (if any, and for reference):

Reviewer's Responses to Questions

**Part I - Summary**

Reviewer #1: The authors have adequately addressed all of the reviewers' comments.

Reviewer #2: Overall, the authors have adequately addressed the concerns.

Reviewer #3: I have no comments to the revised version of this manuscript. This revision was thoroughly prepared and addresses all of the previous comments and suggestions.

**Part II – Major Issues: Key Experiments Required for Acceptance**

Reviewer #1: (No Response)

Reviewer #2: (No Response)

Reviewer #3: No additional experiments are required.

**Part III – Minor Issues: Editorial and Data Presentation Modifications**

Reviewer #1: There needs to be a legend for the supplemental movie, which I had to watch 3 times to see the flip. Labeling would be very helpful.

Reviewer #2: (No Response)

Reviewer #3: No further comments.

PLOS authors have the option to publish the peer review history of their article (what does this mean?). If published, this will include your full peer review and any attached files.

Reviewer #1: **Yes: **Esther Bullitt

Reviewer #2: No

Reviewer #3: **Yes: **Konstantin V. Korotkov

---

## [Editor Report · Acceptance letter]

31 Mar 2021

Dear Dr. Sokurenko,

We are delighted to inform you that your manuscript, "Toggle switch residues control allosteric transitions in bacterial adhesins by participating in a concerted repacking of the protein core.," has been formally accepted for publication in PLOS Pathogens.

Best regards,

Kasturi Haldar

Editor-in-Chief

PLOS Pathogens

orcid.org/0000-0001-5065-158X

Michael Malim

Editor-in-Chief

PLOS Pathogens

orcid.org/0000-0002-7699-2064